# Reinstatement of long-term memory following erasure of its behavioral and synaptic expression in *Aplysia*

**Shanping Chen[1][†], Diancai Cai[1][†], Kaycey Pearce[1], Philip Y-W Sun[1][‡], Adam C Roberts[1], David L Glanzman[1,2,3]\***

[1]Department of Integrative Biology and Physiology, University of California, Los Angeles, Los Angeles, United States; [2]Department of Neurobiology, David Geffen School of Medicine at UCLA, Los Angeles, United States; [3]Integrative Center for Learning and Memory, Brain Research Institute, University of California, Los Angeles, Los Angeles, United States

**Abstract** Long-term memory (LTM) is believed to be stored in the brain as changes in synaptic connections. Here, we show that LTM storage and synaptic change can be dissociated. Cocultures of *Aplysia* sensory and motor neurons were trained with spaced pulses of serotonin, which induces long-term facilitation. Serotonin (5HT) triggered growth of new presynaptic varicosities, a synaptic mechanism of long-term sensitization. Following 5HT training, two antimnemonic treatments—reconsolidation blockade and inhibition of PKM—caused the number of presynaptic varicosities to revert to the original, pretraining value. Surprisingly, the final synaptic structure was not achieved by targeted retraction of the 5HT-induced varicosities but, rather, by an apparently arbitrary retraction of both 5HT-induced and original synapses. In addition, we find evidence that the LTM for sensitization persists covertly after its apparent elimination by the same antimnemonic treatments that erase learning-related synaptic growth. These results challenge the idea that stable synapses store long-term memories.

**\*For correspondence:**
dglanzman@physci.ucla.edu

[†]These authors contributed equally to this work

**Present address:** [‡]Mayo Medical School, Rochester, United States

**Competing interests:** The authors declare that no competing interests exist.

## Introduction

There is significant empirical support for the idea, proposed by Ramón Y Cajal more than a century ago (*Cajal, 1894*), that long-term memories are expressed in the brain, in part, by changes in synaptic connectivity. A corollary of this idea, accepted by many, if not most, modern neuroscientists, is that memories are maintained by persistent molecular and cellular alterations in synaptic structures themselves (*Bailey and Kandel, 2008*; *Kandel et al., 2014*). Here, we have tested the idea that long-term memory (LTM) is stored at synapses using the marine mollusk *Aplysia californica*. The relatively simple behavior and nervous system of this model invertebrate organism offer several major advantages for understanding memory at the level of modifications of individual synapses (*Kandel, 2001*). A form of learning in *Aplysia* whose cellular and molecular substrates are particularly well understood is sensitization of the gill- and siphon-withdrawal reflex (*Carew et al., 1971*; *Brunelli et al., 1976*; *Antonov et al., 1999*; *Kandel, 2001*; *Glanzman, 2010*). Sensitization of the withdrawal reflex exhibits a long-term (≥24 hr) form (*Pinsker et al., 1973*) due, in part, to long-term facilitation (LTF) of the monosynaptic connection between the sensory and motor neurons that mediate the reflex (*Frost et al., 1985*). Importantly, the monosynaptic sensorimotor connection can be reconstituted in dissociated cell culture, and LTF of the in vitro synapse can be induced by training with pulses of serotonin (5HT), the monoaminergic neurotransmitter that mediates sensitization in *Aplysia* (*Brunelli et al., 1976*; *Glanzman et al., 1989*; *Marinesco and Carew, 2002*). Cellular and molecular analyses of this form of long-term

**eLife digest** Cells called neurons allow information to travel quickly around the body so that we can rapidly respond to any changes that we sense in our environment. This includes non-conscious reactions, such as the knee-jerk reflex in humans.

Reflexes and other behaviors can be influenced by long-term memory, and it is thought that long-term memory is stored by changes in the synapses that connect neurons to each other. The reflexes of a sea slug known as *Aplysia* are often used to study memory because it has a simple nervous system in which individual sensory neurons (which detect changes) only form synapses with single motor neurons (which control muscles).

Chen et al. have now studied whether long-term memory is actually stored in these synapses. Sensory neurons and motor neurons removed from *Aplysia* were grown together in Petri dishes and allowed to form synapses. Next, the cells were treated with the hormone serotonin, which promotes long-term memory by, in part, causing the neurons to grow more synapses.

Afterwards, the cells were given treatments that disrupted long-term memory and also reversed the synaptic growth caused by serotonin. However, it was not only new synapses that retracted: some synapses that had existed before the serotonin treatment were also lost. This apparently random loss of synapses suggests that the memory was not stored in specific synapses. Moreover, long-term memory could be restored after these treatments, which supports that idea that memory does not depend on synapses between the neurons being maintained.

This work offers hope that it might be possible to develop treatments that help to restore long-term memory in people suffering from Alzheimer's disease and other conditions that affect long-term memory.

synaptic plasticity have provided major mechanistic insights into long-term memory in *Aplysia* (*Goelet et al., 1986*; *Dash et al., 1990*; *Bartsch et al., 1995*; *Martin et al., 1997*), insights that have generalized to learning and memory in other organisms, including mammals (*Yin et al., 1994*, *1995*; *Frey and Morris, 1997*; *Kogan et al., 1997*; *Josselyn et al., 2001*). Accordingly, we used the in vitro sensorimotor synapse in initial experiments to determine whether LTM is stored at synapses.

Current evidence supports the idea that LTM can be modified or even eliminated under certain circumstances. One of these goes under the rubric of reconsolidation blockade. Here, a stimulus is delivered to an animal that serves to reactivate the LTM for a previous learning experience. If, immediately after delivery of this reminder, the animal is treated with an inhibitor of protein synthesis (*Nader et al., 2000*), or subjected to electroconvulsive shock (*Misanin et al., 1968*), the LTM will be apparently eliminated. Based on this evidence, it has been proposed that the reminder stimulus returns the LTM to a labile state in which new protein synthesis is required to reconsolidate the memory; and that inhibition of protein synthesis during this period of reconsolidation can erase the original memory (*Nader and Hardt, 2009*) (but see *Lattal and Abel, 2004*). Another manipulation that can apparently erase LTM permanently is inhibition of the constitutively active catalytic fragment of the atypical protein kinase Cζ; the ongoing activity of this catalytic fragment, named PKMζ, appears to be required for the maintenance of several forms of LTM in mammals (*Sacktor, 2011*); inhibition of PKMζ, in the absence of a reminder, has been reported to abolish consolidated memory (but see *Lee et al., 2013*; *Volk et al., 2013*).

We, and others, have shown that the synaptic memory for LTF of in vitro sensorimotor connections can be apparently eliminated when sensorimotor cocultures are treated with a protein synthesis inhibitor following reminder training (*Cai et al., 2012*; *Lee et al., 2012*; *Hu and Schacher, 2014*). In addition, inhibiting the activity of PKM Apl III, the constitutively active fragment of the *Aplysia* atypical protein kinase C (PKC Apl III) (*Villareal et al., 2009*; *Cai et al., 2011*; *Bougie et al., 2012*), also erases consolidated LTF (*Cai et al., 2011*). LTF is mediated partly by the growth of new sensory neuron varicosities (*Glanzman et al., 1990*; *Bailey and Kandel, 2008*). Here we investigated whether blocking the reconsolidation of the memory for LTF, or inhibiting PKM Apl III, altered this long-term change in presynaptic structure. We found that the synaptic growth induced by LTF training was reversed by these two memory-disrupting manipulations; however, although the overall number of presynaptic varicosities reverted to the original, pretraining level, the resultant morphological pattern of sensorimotor

synapses differed significantly from the original one. These results imply that the persistence of memory does not require the stability of particular synaptic connections.

We provide additional support for this idea with data from behavioral experiments in which we show that LTM can be reinstated in intact *Aplysia* following its apparent disappearance due to reconsolidation blockade or PKM inhibition. Because these two antimnemonic manipulations not only disrupt the behavioral expression of LTM, but also eliminate the synaptic changes—both electrophysiological (*Cai et al., 2011*, *2012*; *Lee et al., 2012*) and morphological (present data)—closely associated with LTM in *Aplysia* (*Bailey and Chen, 1983*, *1988*; *Frost et al., 1985*; *Glanzman et al., 1990*), our results challenge the idea that the synapse is a cellular site for long-term memory storage in *Aplysia*. In other behavioral experiments we show that both the disruption of LTM through inhibition of PKM Apl III and LTM induction require epigenetic changes. These results point to the nucleus of neurons as the potential locus of the engram in *Aplysia*.

## Results

### Effect of blockade of reconsolidation of long-term synaptic memory on synaptic structure

In electrophysiological experiments involving 5HT-induced LTF of sensorimotor synapses in dissociated cell culture, we previously showed that a 'reminder' stimulus—a single, 5-min pulse of 5HT—could trigger the apparent reconsolidation of synaptic memory, as indicated by the vulnerability of consolidated LTF to disruption by administration of a protein synthesis inhibitor (anisomycin) following the reminder. Specifically, treatment with a single pulse of 5HT and anisomycin 24 hr or more after the induction of LTF reversed the synaptic facilitation (*Cai et al., 2012*; see also; *Lee et al., 2012*; *Hu and Schacher, 2014*). Here we asked whether the blockade of the reconsolidation of LTF also reverses the synaptic growth that underlies LTF (*Glanzman et al., 1990*).

Sensory neurons (SNs) of established sensorimotor cocultures were labeled with dextran fluorescein, and motor neurons (MNs) with dextran rhodamine, via pressure injection; the neurons were then imaged using laser scanning confocal fluorescence microscopy, and the presynaptic varicosities in contact with a postsynaptic structure (either a motor neurite or the postsynaptic soma) were quantified (*Glanzman et al., 1990*) (*Figure 1A,B*). After initial imaging some cocultures received five spaced 5-min pulses of serotonin (5X5HT training, 100 μM). 24 hr later the cocultures were reimaged and the varicosities requantified. As previously reported (*Glanzman et al., 1990*), the 5X5HT training produced a significant increase in the number of presynaptic varicosities contacting the MN (*Figure 1C*). After reimaging, two groups of the trained cocultures received a single 5-min pulse of 5HT (1X5HT, 100 μM) to reactivate the synaptic memory induced by 5X5HT training (*Cai et al., 2012*). Immediately following the reminder pulse of 5HT, one group (5X5HT-1X5HT-Aniso group) received anisomycin (10 μM) treatment for 2 hr. Two other groups of 5X5HT-trained cocultures did not get the reminder pulse of 5HT; one of these groups (5X5HT-Aniso) received the anisomycin treatment, whereas the other group (5X5HT) received vehicle solution instead. A final group of cocultures (Controls) did not receive either 5HT or anisomycin; instead, the Controls were treated with vehicle solution at the experimental times that the 5X5HT training and anisomycin treatment were administered to other groups. At 48 hr after the original imaging session all of the cocultures were imaged and the varicosities once more quantified.

The overall increase in the number of presynaptic varicosities induced by 5X5HT training on Day 1 persisted through the third imaging session in the 5X5HT and 5X5HT-Aniso groups of cocultures (*Figure 1C*). (Notice that because there were no significant differences between the 5X5HT and 5X5HT-1X5HT groups, these two groups have been combined into a single group [5X5HT] in *Figure 1C*. However, the data for the 5X5HT-1X5HT group are presented separately in the graph in *Figure 1—figure supplement 1*). The reminder pulse of 5HT coupled with inhibition of protein synthesis caused the number of varicosities to revert to the pretraining (0 hr) value in the 5X5HT-1X5HT-Aniso group. This structural result parallels the electrophysiological results previously reported for *Aplysia* sensorimotor cocultures (*Cai et al., 2012*; *Lee et al., 2012*; *Hu and Schacher, 2014*), and provides additional support for the notion that 1X5HT reactivated the synaptic memory induced by the 5X5HT training.

Besides quantifying changes in overall varicosity number, we tracked the fate of each SN varicosity in every coculture over the course of the experiments. The varicosities were put into one of three categories:

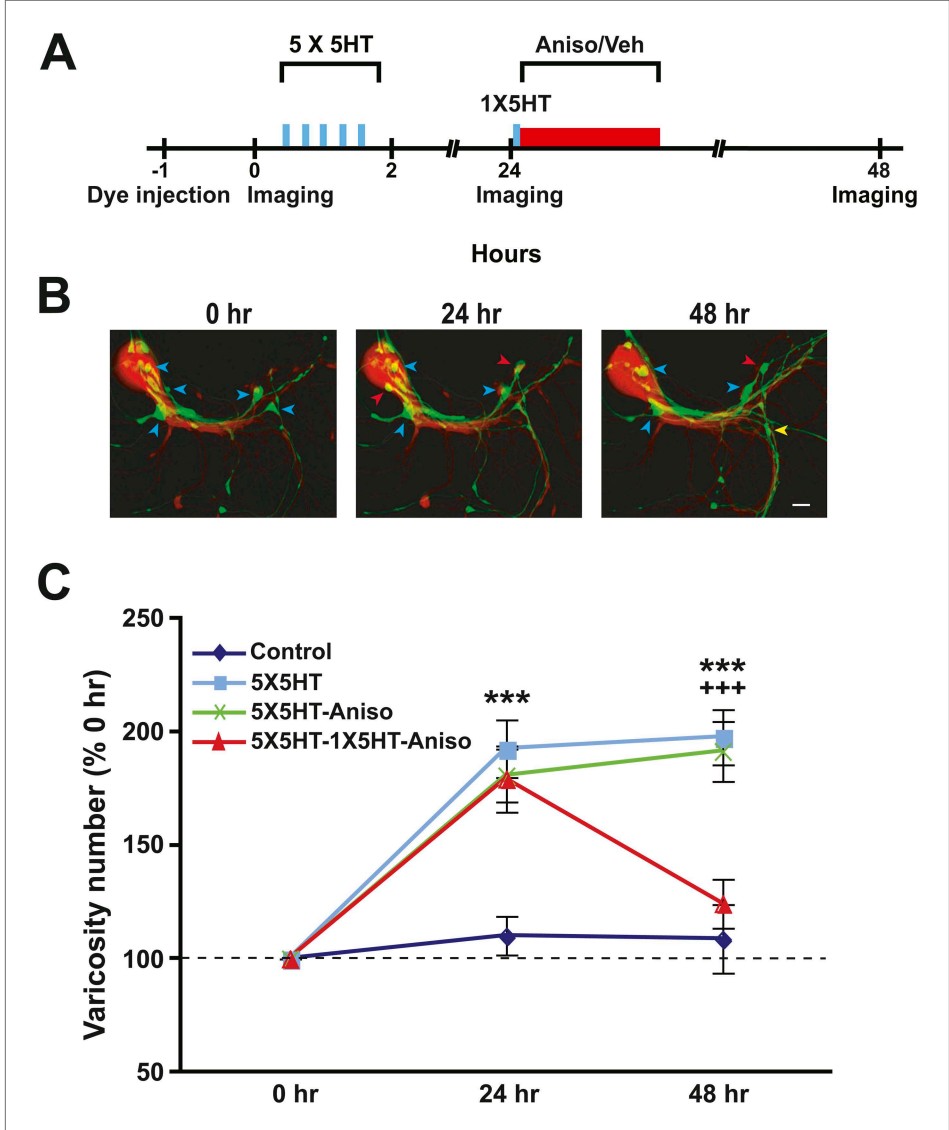

**Figure 1**. Blockade of memory reconsolidation reverses 5HT-induced synaptic growth. (**A**) Experimental protocol. The vertical blue bars represent pulses of 5HT, and the horizontal red bar represents anisomycin/vehicle treatment. A reminder pulse of 5HT (single blue bar) was delivered to the 5X5HT-1X5HT-Aniso cocultures prior to the anisomycin. (**B**) Sample confocal micrographs of a Control coculture. Blue arrowheads, presynaptic varicosities present at 0 hr; red arrowheads, new varicosities that appeared during 0–24 hr; and yellow arrowheads, varicosities formed during the 24–48 hr period. Scale bar, 20 μm. (**C**) Mean normalized varicosity number at 24 hr and 48 hr in the Control (n = 21), 5X5HT (n = 21), 5X5HT-Aniso (n = 19), and 5X5HT-1X5HT-Aniso (n = 26) groups. The number of varicosities measured at 24 hr and 48 hr was normalized to the number present at 0 hr (Note that differences in the data for the 5X5HT and 5X5HT-1X5HT groups were not significant, and two groups have been grouped together in the graph. The separate data for the 5X5HT-1X5HT group can be found in *Figure 1—figure supplement 1*). A repeated-measures ANOVA indicated that there was a significant group × time interaction ($F_{[3,83]}$ = 9.5, p < 0.0001). Planned comparisons using one-way ANOVAs indicated that the group differences at 24 and 48 hr were highly significant ($F_{[3,83]}$ = 8.6 for 24 hr and 12.5 for 48 hr; p < 0.0001 for the results of each ANOVA). SNK posthoc tests on the 24 hr data showed that the mean normalized varicosity number in the 5X5HT group (192.8 ± 12.7%), 5X5HT-Aniso group (180.9 ± 11.6%), and 5X5HT-1X5HT-Aniso group (179.5 ± 14.6%) was each significantly greater than that in the Control group (110.3 ± 8.5%; p < 0.001 for all comparisons). The increase in varicosity number persisted to 48 hr in the 5X5HT group (197.9 ± 12.2%, p < 0.001) and 5X5HT-Aniso group (191.6 ± 13.2%, p < 0.001), but not in the 5X5HT-1X5HT-Aniso group (124.4 ± 10.9%), when compared to the Control group (108.9 ± 15.2%). The difference between the 5X5HT-Aniso and 5X5HT-1X5HT-Aniso groups was highly significant (p < 0.001). *Figure 1. Continued on next page*

*Figure 1. Continued*

Asterisks indicate a significant difference for comparisons with the Control group; plus signs indicate a significant difference for comparisons with the 5X5HT-1X5HT-Aniso group. Here and in subsequent figures one symbol indicates p < 0.05; two symbols, p < 0.01; three symbols, p < 0.001. Error bars represent ±SEM.

The following figure supplement is available for figure 1:

**Figure supplement 1**. Graphs presenting the normalized mean varicosity numbers for the three 5HT-trained groups not subjected to reconsolidation blockade.

---

'original' varicosities, that is, the varicosities present at 0 hr; '5HT-induced' varicosities, the varicosities that appeared during the 24 hr after 5HT treatment (this category pertained only to the groups given 5X5HT training); and 'new' varicosities, varicosities formed during the 24–48 hr period. For the analysis of varicosity fate the three groups of cocultures that received the 5X5HT training without reconsolidation blockade—that is, the 5X5HT, 5X5HT-1X5HT and 5X5HT-Aniso groups—were consolidated into a single group labeled '5HT-No reconsolidation/No blockade' in *Figure 2*. Inspection of the fates of individual varicosities in this group yielded surprising results. First, many of the 5HT-induced SN varicosities in the 5HT-No reconsolidation/No blockade group did not persist until the final imaging session (*Figures 2A and 3A*); instead, there was significant retraction of the 5HT-induced varicosities between 24 and 48 hr in this group. Second, there was also retraction of the original varicosities during this period (*Figures 2B and 3A*). Varicosities in the 5X5HT-1X5HT-Aniso group—the group of trained cocultures subjected to reconsolidation blockade—exhibited a similar pattern of retraction of 5HT-induced and original varicosities, but the amount of retraction was significantly greater than that observed in the 5HT-No reconsolidation/No blockade cocultures (*Figures 2A,B and 3B*). A third unexpected finding was the substantial growth of new varicosities in all trained groups between 24 and 48 hr; the amount of the growth was significantly greater, however, in the 5-HT-No reconsolidation/No blockade cocultures than in the 5X5HT-1X5HT-Aniso group (*Figure 2C*). Thus, the 5X5HT trained cocultures subjected to reconsolidation blockade exhibited significantly more retraction and significantly less growth of varicosities during the 24–48 hr period than did the other trained cocultures. Interestingly, whereas there was greater retraction of 5HT-induced varicosities than of original varicosities between 24–48 hr in the 5HT-No reconsolidation/No blockade group, the retraction of 5HT-induced varicosities and original varicosities did not differ in the 5X5HT-1X5HT-Aniso group (*Figure 2—figure supplement 1*).

The substantial morphological changes in the 5HT-No reconsolidation/No blockade cocultures between 24 and 48 hr were surprising, because the overall number of varicosities remained stable in these cocultures during this period (see the data for the 5X5HT and 5X5HT-Aniso groups in *Figure 1C*). Moreover, as demonstrated in our earlier electrophysiological investigations of LTF, the amplitude of the sensorimotor EPSP was also stable between 24 and 48 hr after 5X5HT training (*Cai et al., 2011*). Our morphological data reveal the operation of a heretofore unrecognized homeostatic mechanism that adjusts the number of SN varicosities—and, presumably, active synaptic sites (*Schacher et al., 1990*; *Kim et al., 2003*)—for sensorimotor connections according to their learning-related experience, but that seems unconcerned with the identities of the individual varicosities/synaptic sites. In the 5HT-No reconsolidation/No blockade cocultures the significant retraction of SN varicosities during 24–48 hr after 5X5HT training was compensated for by significant growth of new varicosities, so that the overall number remained constant (*Figures 1C and 2*). By contrast, the number of varicosities was reset to the original (0 hr) value in the 5X5HT-1X5T-Aniso group; this resetting involved increased retraction of both 5HT-induced and original varicosities, and decreased growth of new varicosities between 24 and 48 hr. Consequently, the morphological state of the cocultures in the 5X5HT-1X5T-Aniso group at 48 hr differed significantly from that at 0 hr with respect to the identities of individual SN varicosities.

## Synaptic structure following inhibition of PKM

Next we examined the effect of inhibiting PKM on SN varicosity number. PKM Apl III, a homolog of mammalian PKMζ (*Sacktor, 2011*), is formed from the atypical *Aplysia* protein kinase C (PKC Apl III) (*Bougie et al., 2009*) by calpain-dependent cleavage; furthermore, this cleavage is induced by prolonged treatment with 5HT (*Bougie et al., 2012*). In previous behavioral and electrophysiological investigations we found that inhibition of PKM Apl III, either with the pseudosubstrate sequence of the

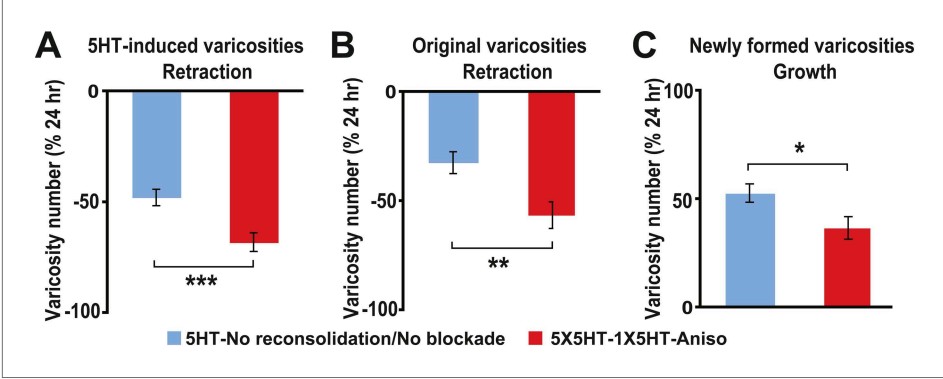

**Figure 2**. Effect of reconsolidation blockade on the fates of individual varicosities. For varicosities in the 5HT-induced and original categories, the varicosity count in a given category for each coculture at 48 hr was normalized to the number of varicosities in the same category in that coculture at 24 hr. For the varicosities formed >24 hr, the varicosity count for each coculture at 48 hr was normalized to the total number of varicosities in that coculture at 24 hr. (**A**) The mean normalized decrease in the number of 5HT-induced varicosities (varicosities that appeared within 24 hr after 5X5HT training) at 48 hr was 67.9 ± 4.2% in the 5X5HT-1X5HT-Aniso group, which was greater than that in the 5HT-No reconsolidation/No blockade group (47.7 ± 3.7%, p < 0.001, two-tailed *t* test). (**B**) Retraction of original varicosities in the 5X5HT-1X5HT-Aniso group (56.4 ± 6.1%) was significantly greater than in the 5HT-No reconsolidation/No blockade group (32.3 ± 5.1%, p < 0.01, two-tailed *t* test). (**C**) Reversal of the morphological changes induced by 5X5HT training also involved inhibition of synaptic growth from 24–48 hr. There was significantly less growth of new varicosities during this period in the 5X5HT-1X5HT-Aniso group (36.8 ± 5.2%) than in the 5HT-No reconsolidation/No blockade group (52.9 ± 4.2%, p < 0.05, two-tailed *t* test). Blue bars, 5HT-No reconsolidation/No blockade; red bars, 5X5HT-1X5HT-Aniso group. Asterisks indicate significant differences between the 5HT-No reconsolidation/No blockade and 5X5HT-1X5HT-Aniso groups. Error bars represent ±SEM.

The following figure supplement is available for figure 2:

**Figure supplement 1**. Comparison of retraction of original and 5HT-induced varicosities between 24–48 hr in the reconsolidation blockade experiments.

regulatory domain of atypical PKC (ZIP) or with chelerythrine, a PKC inhibitor selective for PKM at low concentrations (***Ling et al., 2002***; ***Villareal et al., 2009***), abolished both consolidated long-term sensitization and LTF (***Cai et al., 2011***). To determine whether inhibition of PKM Apl III reverses the structural growth that mediates long-term sensitization (***Bailey and Chen, 1983***, ***1988***) and LTF (***Glanzman et al., 1990***), we gave some cocultures (5X5HT-Chel group) 5X5HT training followed 24 hr later by treatment with chelerythrine (***Figure 4A***). Another group of cocultures (5X5HT group) received the 5X5HT training alone. Finally, a third group (Controls) received the vehicle solution instead of either 5HT or chelerythrine. The SNs and MNs of all of the cocultures were labeled with fluorescent dye as before and then imaged on Day 1 (0 hr) of the experiment. Immediately afterwards cocultures in the 5X5HT and 5X5HT-Chel groups were given 5X5HT training. All cocultures were imaged for a second time at 24 hr, after which cocultures in the 5X5HT-Chel group received chelerythrine (10 µM, 1 hr), while the other two groups received the vehicle solution. (The 5X5HT-Chel group did not receive 1X5HT prior to chelerythrine treatment.) The cocultures were imaged for a final time at 48 hr.

As before, the 5X5HT training produced a significant net increase in the number of presynaptic varicosities in the two trained groups at 24 hr (***Figure 4B***). This net increase persisted in the 5X5HT group, but was reversed in 5X5HT-Chel group at 48 hr. Monitoring of the fate of individual varicosities in the 5X5HT group revealed the same pattern of structural growth and retraction observed previously in the 5X5HT-trained cocultures not subjected to reconsolidation blockade. Specifically, some varicosities induced by the 5X5HT training were lost between 24 and 48 hr, as were some of the original varicosities; this loss, however, was compensated for by the growth of new varicosities; consequently, the overall varicosity number remained stably elevated during the 24–48 hr period (***Figures 3C and 5***). The varicosities in the 5X5HT-Chel group exhibited significantly greater retraction and significantly less growth between 24 and 48 hr than those in the 5X5HT group, resulting in an overall loss of varicosities. Thus, the number of varicosities in the 5X5HT-Chel group at 48 hr was returned to the value at 0 hr.

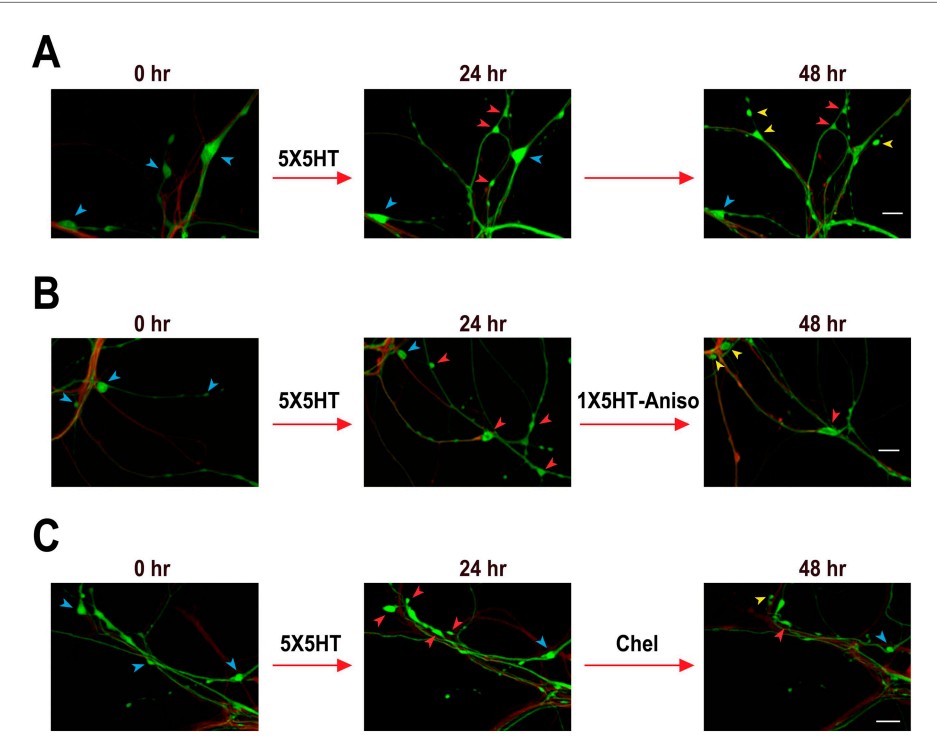

**Figure 3**. Confocal fluorescence micrographs illustrating the structural effects of 5X5HT training, reconsolidation blockade, and chelerythrine treatment. (**A**) Coculture that received 5X5HT training alone. (**B**) Coculture that received the 5X5HT training plus 1X5HT and anisomycin treatment immediately after the 24 hr imaging session. (**C**) Coculture that received the 5X5HT training plus chelerythrine treatment immediately after the 24 hr imaging session. Blue arrowheads, varicosities present at 0 hr; red arrowheads, varicosities formed during 0–24 hr; and yellow arrowheads, varicosities formed during 24–48 hr. Scale bars, 20 µm.

This result indicates that inhibiting PKM Apl III, like reconsolidation blockade (*Figure 1C*), engages a homeostatic mechanism that resets the presynaptic varicosities in sensorimotor cocultures to the number present prior to training. Another similarity between the morphological effects of reconsolidation blockade and inhibition of PKM was that there was equal retraction of 5HT-induced and original varicosities in the 5X5HT-Chel cocultures during 24–48 hr, whereas, as before, the retraction of 5HT-induced varicosities was greater in the 5X5HT cocultures during this period (*Figure 5—figure supplement 1*).

Inspection of the synaptic structure of the Control cocultures revealed a complex underlying pattern of structural change similar to that observed in the reconsolidation experiment. There was significant growth and retraction of SN varicosities, both original and new, over the 48 hr of the experiments (*Figures 6 and 1B*). This structural dynamism contrasted with the constancy of overall varicosity number in these cocultures (*Figures 1C and 4B*), as well as with the stability of the sensorimotor EPSP amplitude observed in Control cocultures over similar time periods in our earlier studies (*Cai et al., 2011, 2012*).

## Additional training can reinstate LTM after its apparent elimination

The present morphological results challenge the notion that the persistence of sensitization memory in *Aplysia* depends on the persistence of particular facilitated synapses. To further investigate this idea, we tested whether the LTM for behavioral sensitization can be reinstated in *Aplysia* following reconsolidation blockade and inhibition of PKM, two treatments previously shown to eliminate LTF, the synaptic basis of long-term sensitization (*Cai et al., 2011, 2012; Lee et al., 2012; Hu and Schacher, 2014*).

Long-term sensitization (LTS) of the siphon-withdrawal reflex (SWR) was induced in intact *Aplysia* using five bouts of tails shocks (5XTrained). Brief reminder training (one bout of tail shocks, 1XTrained)

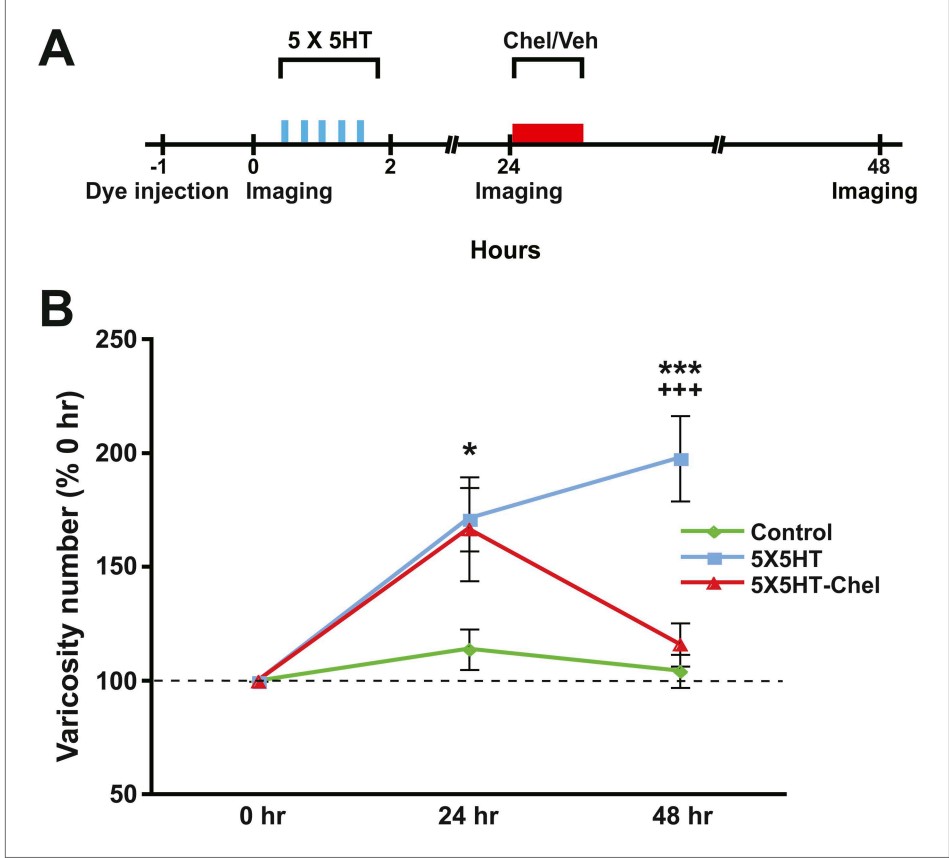

**Figure 4**. Inhibition of PKM also reverses 5HT-induced synaptic growth. (**A**) Experimental protocol. The vertical blue bars represent pulses of 5HT; the horizontal red bar represents the period of chelerythrine/vehicle treatment. The 5X5HT and Control groups received vehicle treatment during this time. (**B**) Mean SN varicosity number measured at 24 hr and 48 hr for the Control ($n = 14$), 5X5HT ($n = 21$) and 5X5HT-Chel ($n = 14$). The number of varicosities measured at 24 hr and 48 hr was normalized to the number of varicosities at 0 hr. A repeated-measures ANOVA indicated that there was a significant group × time interaction ($F_{[2,46]} = 7.5$, $p < 0.001$). Planned comparisons showed that the group differences were significant for both the 24 and 48 hr measurements (24 hr, $F_{[2,46]} = 3.7$, $p < 0.04$; 48 hr, $F_{[2,46]} = 12.8$, $p < 0.0001$). The mean normalized varicosity number at 24 hr was 171.2 ± 14.0% in the 5X5HT group, 166.9 ± 22.8% in the 5X5HT-Chel group, and 114.0 ± 8.9% in the Control group. Posthoc comparisons indicated that the 5X5HT training produced a significant increase in varicosity number in the 5X5HT and 5X5HT-Chel groups at 24 hr ($p < 0.05$ for both comparisons with the Control group). The mean normalized varicosity number at 48 hr in the 5X5HT group (197.9 ± 18.7%) was greater than that in the Control group (104.4 ± 7.2%, $p < 0.001$), but the 5X5HT-Chel group mean (116.1 ± 9.5%) was not significantly different from the Control mean. The difference between the mean varicosity numbers for the 5X5HT and 5X5HT-Chel groups was highly significant ($p < 0.001$). Asterisks, comparison between 5X5HT and Control groups; plus signs, comparison between 5X5HT and 5X5HT-Chel groups. Error bars represent ±SEM.

was applied at 48 hr after the original training to trigger reconsolidation of the LTM for sensitization (*Cai et al., 2012*). Immediately following the reminder training an intrahemocoel injection of anisomycin was administered to some animals (*Figure 7A*). As previously reported (*Cai et al., 2012*; *Lee et al., 2012*), this treatment eliminated LTS, assessed here at 72 hr (*Figure 7B*). Afterwards, some of the animals received three additional bouts of tail shocks (3XTrained). Importantly, the three additional bouts of sensitization training did not induce LTM in naïve animals (Control-Veh-3XTrained; *Figure 7B*). However, the three bouts of training completely restored LTM following its disruption by reconsolidation blockade (*Figure 7B*).

To test whether LTS could be reinstated after its apparent erasure by inhibition of PKM Apl III, animals were again initially trained using five bouts of tail shocks. 24 hr after the original sensitization training the animals received an intrahemocoel injection of chelerythrine (*Figure 8A*). Animals treated

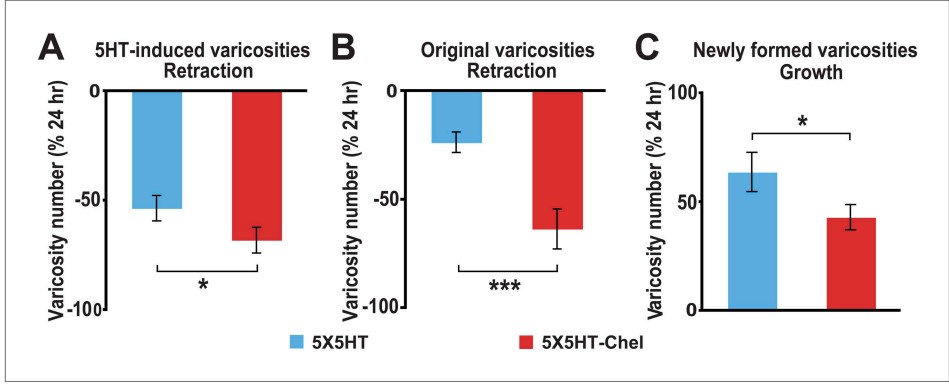

**Figure 5**. Effect of inhibition of PKM on varicosity fate. For the 5HT-induced and original varicosities, the number of varicosities in each category for each coculture at 48 hr was normalized to the number in the same category in that coculture at 24 hr. For the varicosities formed >24 hr, the number of varicosities for each coculture at 48 hr was normalized to the total number of varicosities in the coculture at 24 hr. (**A**) The mean normalized decrease in the number of 5HT-induced varicosities at 48 hr was 53.4 ± 5.8% in the 5X5HT group and 68.0 ± 5.8% in the 5X5HT-Chel group (p < 0.05, one tailed *t* test). (**B**) The mean normalized decrease in the number of original varicosities at 48 hr was 23.5 ± 4.7% in the 5X5HT group and 63.4 ± 9.3% in the 5X5HT-Chel group (p < 0.001, two-tailed *t* test). (**C**) More presynaptic varicosities were formed during 24–48 hr in the 5X5HT group (63.9 ± 9.0%) than in the 5X5HT-Chel group (43.2 ± 5.8%, p < 0.05, one-tailed *t* test). Blue bars, 5X5HT group; red bars, 5X5HT-Chel group. Asterisks, comparison between 5X5HT and 5X5HT-Chel groups. Error bars represent ±SEM.

The following figure supplement is available for figure 5:

**Figure supplement 1**. Comparison of retraction of original and 5HT-induced varicosities between 24–48 hr in the chelerythrine treatment experiments.

with chelerythrine exhibited no LTS at 48 hr (*Figure 8B*), confirming our previous finding (*Cai et al., 2011*). The animals that received the original sensitization training were then given three additional bouts of tail shocks (5XTrained-Chel-3XTrained group), as were control animals that had not received the original sensitization training (Control-Veh-3XTrained group). The modest additional training reinstated LTS following its apparent erasure by chelerythrine, but did not produce LTS in the control animals (*Figure 8B*).

## Role of histone deacetylase in the maintenance and induction of LTM

The above behavioral results indicate that LTM in *Aplysia* can persist despite elimination of its behavioral expression by reconsolidation blockade and inhibition of PKM Apl III. Moreover, because these two antimnemonic treatments also eliminate LTF (*Cai et al., 2011*, *2012*; *Lee et al., 2012*; *Hu and Schacher, 2014*) (and *Figures 1C and 4B*), the results imply that some component of LTM, perhaps a priming process, may persist in the absence of synaptic alterations. Possibly, LTM, or the primer for LTM, resides in the nuclei of neurons within the SWR circuitry, encoded as epigenetic changes (*Levenson and Sweatt, 2005*; *Rahn et al., 2013*; *Graff and Tsai, 2013a*). To investigate this possibility, we tested whether chelerythrine's apparent disruption of LTM involves alterations of chromatin structure. For this test we used the histone deacetylase (HDAC) inhibitor trichostatin A (TSA) (*Graff and Tsai, 2013b*). *Aplysia* were given five bouts of tail shocks, and the strength of the SWR was tested 24 hr later (*Figure 9A*). Immediately after this test the sensitization-trained animals received an intra-hemocoel injection of the vehicle solution alone or TSA; 10–15 min later some of the animals treated with TSA were given an injection of chelerythrine (5XTrained-TSA-Chel group). (Untrained control animals received an injection of the vehicle solution after the 24 hr test.) In addition, another group of trained animals were given the chelerythrine without a prior injection of TSA (5XTrained-Chel group). All animals were retested at 48 hr. TSA treatment blocked the disruption of LTS by chelerythrine (*Figure 9B*). The injection of TSA 24 hr after training by itself did not alter LTS (5XTrained-TSA group). These results indicate that one of chelerythrine's antimnemonic actions is to reverse histone acetylation induced by LTS training.

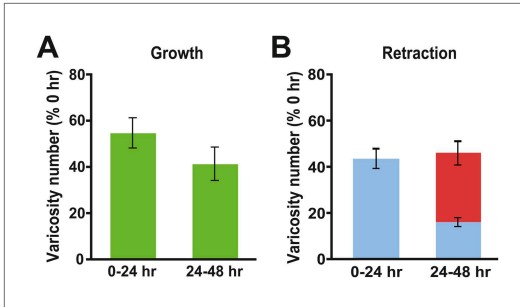

**Figure 6.** Changes in presynaptic varicosities in Control cocultures. (**A**) The normalized increase in the number of varicosities was 54.9 ± 6.5% during the 0–24 hr period and 41.5 ± 7.2% during the 24–48 hr period (green bars). (**B**) The mean normalized number of original varicosities (blue bar) that retracted during the 0–24 hr period was 43.2 ± 4.4%. During the 24–48 hr period 15.5 ± 2.0% of the original varicosities (those present at 0 hr, blue segment), and 30.2 ± 5.1% of the varicosities that formed during 0–24 hr period (red segment), retracted; thus, the mean total normalized retraction of varicosities from 24–48 hr was 45.7%. Note that the numbers of newly formed varicosities during the 0–24 and 24–48 hr periods in each coculture were normalized to the total number of varicosities in that coculture measured at 0 hr. The percentage of retracted varicosities was obtained by normalizing the number of varicosities that disappeared during the 0–24 and 24–48 hr periods for a given coculture to the total number of varicosities in that coculture at 0 hr.

Previous work has shown that LTF of in vitro *Aplysia* sensorimotor connections depends on alterations of chromatin structure. In particular, 1X5HT, which normally induces only short-term facilitation, was found to induce LTF when applied with TSA (*Guan et al., 2002*). In intact *Aplysia* three bouts of tail shocks are insufficient by themselves to induce LTS (Control-Veh-3XTrained data, *Figures 7 and 8*). We tested whether three bouts of shocks could induce LTS if delivered in the presence of an HDAC inhibitor. Animals were given three bouts of shocks 15 min after an intra-hemocoel injection of either vehicle solution or TSA (*Figure 9C*). 24 hr later the animals that received three bouts of shocks after an injection of TSA exhibited LTS, whereas animals that received the shocks after an injection of vehicle did not (*Figure 9D*). Thus, HDAC inhibition facilitates the induction of LTM in *Aplysia*.

## Discussion

The present morphological results, together with those of our previous behavioral and electrophysiological investigations (*Cai et al., 2011*, *2012*), suggest that the persistence of sensitization-related LTM in *Aplysia* does not require the persistence of the synaptic connections generated during learning. Rather, LTM appears to be regulated by a homeostatic mechanism that specifies the net synaptic strength according to experience. Following 5X5HT stimulation, we observed that the number of presynaptic varicosities in the sensorimotor cocultures increased to a new overall value, and that this value was maintained despite the appearance and disappearance of individual varicosities. Furthermore, blockade of memory reconsolidation, or inhibition of PKM, reset the number of varicosities to the original, nonfacilitated value. Remarkably, this resetting did not result from the straightforward retraction of the varicosities that appeared by 24 hr after 5X5HT stimulation, varicosities whose growth coincides with LTF (*Glanzman et al., 1990*); rather, the resetting was produced by a complex reorganization that involved retraction of both 5HT-induced and original varicosities, as well as growth of new, additional varicosities. Thus, for any given sensorimotor connection, the homeostatic regulatory mechanism appeared indifferent to the identities of the particular presynaptic varicosities and, presumably, active zones (*Schacher et al., 1990*), that mediated baseline synaptic transmission and synaptic facilitation. Our data argue that synapses are not 'tagged' with respect to memory storage, at least for a connection involving a single SN and a single MN; the data may therefore reflect a fundamental distinction between the cellular and molecular processes of LTM storage and those of LTM induction in *Aplysia*, in which synaptic tagging plays a critical role (*Martin et al., 1997*). The data also suggest that synaptic change is an expression mechanism, rather than a storage mechanism, for LTM in *Aplysia* (although see below).

Importantly, neither reconsolidation blockade nor inhibition of PKM caused the varicosity number to drop below the starting value. (Notice that the same is true for the amplitude of the EPSP in our earlier synaptic physiological studies, which used the same training regimens [*Cai et al., 2011*, *2012*].) This result provides additional evidence for the operation of a homeostatic regulatory mechanism, and suggests that the homeostatic mechanism toggles between two all-or-none states, a facilitated, sensitization-related state and a nonfacilitated one. (There must also be a third state, synaptic depression, in which the strength of the sensorimotor connection is reduced below the nonfacilitated level [*Montarolo et al., 1988*].) Our results are reminiscent of those from studies in the *Drosophila*

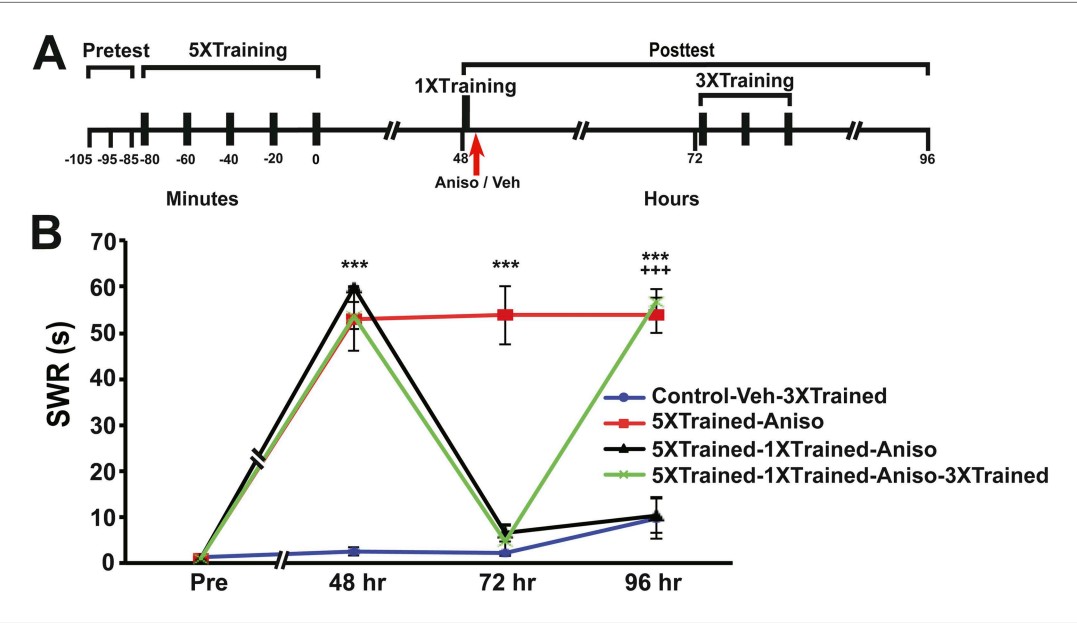

**Figure 7**. Reinstatement of LTS after its elimination by reconsolidation blockade in *Aplysia*. (**A**) Experimental protocol. The timing of the pretests, training, posttests, and drug/vehicle injections is shown relative to the end of the last training session. The time of the intrahemocoel injection of either anisomycin or vehicle is indicated by the red arrow. Animals in the 5XTrained-1XTrained-Aniso and 5XTrained-1XTrained-Aniso-3XTrained groups received a reminder episode of sensitization training (one bout of tail shocks) immediately after the 48 hr posttest (black bar) and prior to the injection of anisomycin, whereas the animals in the 5XTrained-Aniso group did not. After the 72 hr posttest animals in the Control-Veh-3XTrained and 5XTrained-1X5HT-Aniso-3XTrained groups received truncated sensitization training (three bouts of tail shocks). (**B**) The mean duration of the SWR measured at 48 hr, 72 hr and 96 hr for the Control-Veh-3XTrained (*n* = 12), 5XTrained-Aniso (*n* = 4), 5XTrained-1XTrained-Aniso (*n* = 7), and 5XTrained-1XTrained-Aniso-3XTrained (*n* = 11) groups. A repeated-measures ANOVA indicated that there was a significant group × time interaction ($F_{[9,90]}$ = 51.0, p < 0.0001). Subsequent planned comparisons indicated that the overall differences among the four groups were highly significant on all of the posttests (48 hr, $F_{[3,30]}$ = 137.2, p < 0.0001; 72 hr, $F_{[3,30]}$ = 50.5, p < 0.0001; and 96 hr, $F_{[3,30]}$ = 43.8, p < 0.0001). SNK posthoc tests on the 48 hr data indicated that the initial sensitization training produced significant LTS in the 5XTrained-Aniso (53.0 ± 7.0 s), 5XTrained-1XTrained-Aniso (59.9 ± 0.2 s), and 5XTrained-1XTrained-Aniso-3XTrained (53.7 ± 2.9 s) groups compared to the Control-Veh-3XTrained group (2.5 ± 0.9 s, p < 0.001 for each test). The responses of the trained groups did not differ significantly at 48 hr after sensitization training. However, the mean duration of the SWR in the 5XTrained-Aniso group (53.8 ± 6.3 s) remained prolonged at 72 hr, and was significantly longer than that in the 5XTrained-1XTrained-Aniso group (6.4 ± 1.9 s) as well as in the 5XTrained-1XTrained-Aniso-3XTrained group (4.8 ± 3.2 s, p < 0.001 for both comparisons). LTS was restored by the three additional tail shocks applied after the 72 hr posttest. The mean duration of the SWR in the 5XTrained-1XTrained-Aniso-3XTrained group at 96 hr was 56.7 ± 2.7 s, which was significantly longer than that for the 5XTrained-1XTrained-Aniso group (10.4 ± 3.9 s) at 96 hr (p < 0.001). The SWR of the 5XTrained-Aniso group at 96 hr (53.8 ± 3.8 s) was also significantly longer than that in the 5XTrained-1XTrained-Aniso group (p < 0.001). Asterisks, comparisons of the Control-Veh-3XTrained group with the 5XTrained-Aniso group, the 5XTrained-1XTrained-Aniso group, and 5XTrained-1XTrained-Aniso-3XTrained group at 48 hr; comparison of the Control-Veh-3XTrained group with the 5XTrained-Aniso group at 72 hr; and comparisons of the Control-Veh-3XTrained group with the 5XTrained-Aniso group and the 5XTrained-1XTrained-Aniso-3XTrained group at 96 hr. Plus signs, comparisons of the 5XTrained-1XTrained-Aniso group with the 5XTrained-Aniso group and the 5XTrained-1XTrained-Aniso-3XTrained group at 96 hr. Error bars represent ±SEM.

neuromuscular junction and the mammalian CNS, which also indicate that synaptic plasticity is regulated by homeostatic mechanisms (*Turrigiano, 2007*).

The present conclusions appear to conflict with those from morphological investigations of LTM in mice in which dendrites have been chronically imaged in the intact, living brain. Several such studies have reported stability of some dendritic spines for periods of weeks-to-months after training (*Xu et al., 2009*; *Yang et al., 2009*; *Liston et al., 2013*). These results, as well as evidence that spines in the adult

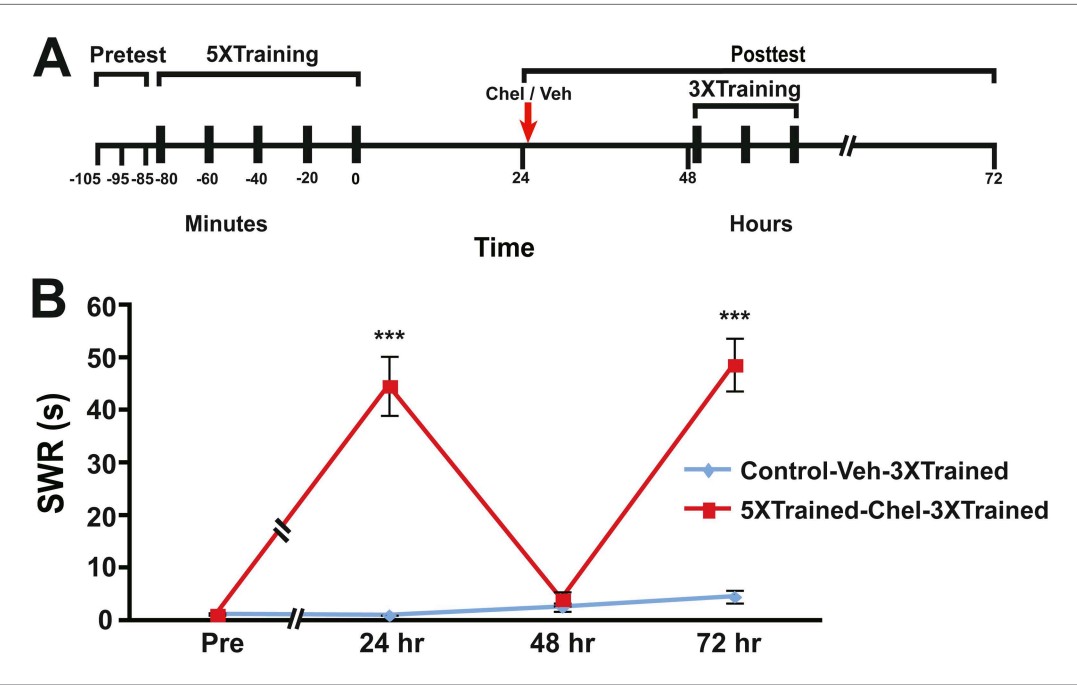

**Figure 8**. Reinstatement of LTS following its erasure by PKM inhibition. (**A**) Experimental protocol. The timing of the pretests, training, posttests, and drug/vehicle injections is shown relative to the end of the last training session. The time of the intrahemocoel injection of either chelerythrine or vehicle is indicated by the red arrow. After the 48 hr posttest animals in both groups received three additional bouts of tail shocks. (**B**) Mean duration of the SWR measured at 24 hr, 48 hr and 72 hr for the Control-Veh-3XTrained ($n = 9$) and 5XTrained-Chel-3XTrained ($n = 9$) groups. A repeated-measures ANOVA indicated that there was a significant interaction effect between group and time ($F_{[3,14]} = 25.4$. $p < 0.0001$). A planned comparison indicated that the mean duration of the SWR at 24 hr in the 5XTrained-Chel-3XTrained group (44.7 ± 5.6 s) was significantly longer than that in the Control-Veh-3XTrained group (1.1 ± 0.1 s, $F_{[1,16]} = 60.2$, $p < 0.001$). The mean duration of the SWR at 48 hr (4.1 ± 1.4 s) in the 5XTrained-Chel-3XTrained group was not significantly longer than that in the Control-Veh-3XTrained group (2.6 ± 0.8 s), as indicated by a planned comparison ($F_{[1,16]} = 0.95$, $p = 0.34$). The SWR in the 5XTrained-Chel-3XTrained group at 72 hr was 48.7 ± 5.0 s, which was significantly longer than that in the Control-Veh-3XTrained group (4.6 ± 1.2 s; planned comparison, $F_{[1,16]} = 73.1$, $p < 0.001$). Asterisks, comparison between 5XTrained-Chel-3XTrained and Control-Veh-3XTrained groups. Error bars represent ±SEM.

mammalian cortex exhibit little turnover (*Grutzendler et al., 2002*), have led some to argue that stable spines provide a physical basis for the lifelong maintenance of memories (*Bhatt et al., 2009*). But other studies of the mouse brain using very similar in vivo imaging techniques have reported a high degree of spine instability in the adult cerebral cortex (*Trachtenberg et al., 2002*; *Holtmaat et al., 2005*); the contradiction between these two sets of opposed findings remains unresolved. Furthermore, even in those studies where spines of adult cortical neurons were reported to be highly stable, there was significant instability of both pre-existing and new spines within the first 2–4 days after a training protocol (*Xu et al., 2009*; *Yang et al., 2009*), and we monitored SN varicosities for only 48 hr after training with 5HT. Therefore, the present results are not necessarily inconsistent with those from in vivo imaging studies in the mouse brain.

We found that LTS could be fully reinstated following its disruption by the same antimnemonic treatments previously shown to eliminate LTF (*Cai et al., 2011*, *2012*; *Lee et al., 2012*; *Hu and Schacher, 2014*) (*Figures 7 and 8*). This finding is consistent with two possible explanations. First, the LTM for sensitization may be intact in animals following reconsolidation blockade and inhibition of PKM, despite the elimination of both behavioral enhancement and synaptic growth. According to this scheme, synapses serve merely to express LTM, they are not sites of LTM storage. The reinstatement of LTM expression, then, involves restoring the appropriate number of synapses between the SNs and MNs, as determined by an experience-registering homeostatic mechanism. The second possible

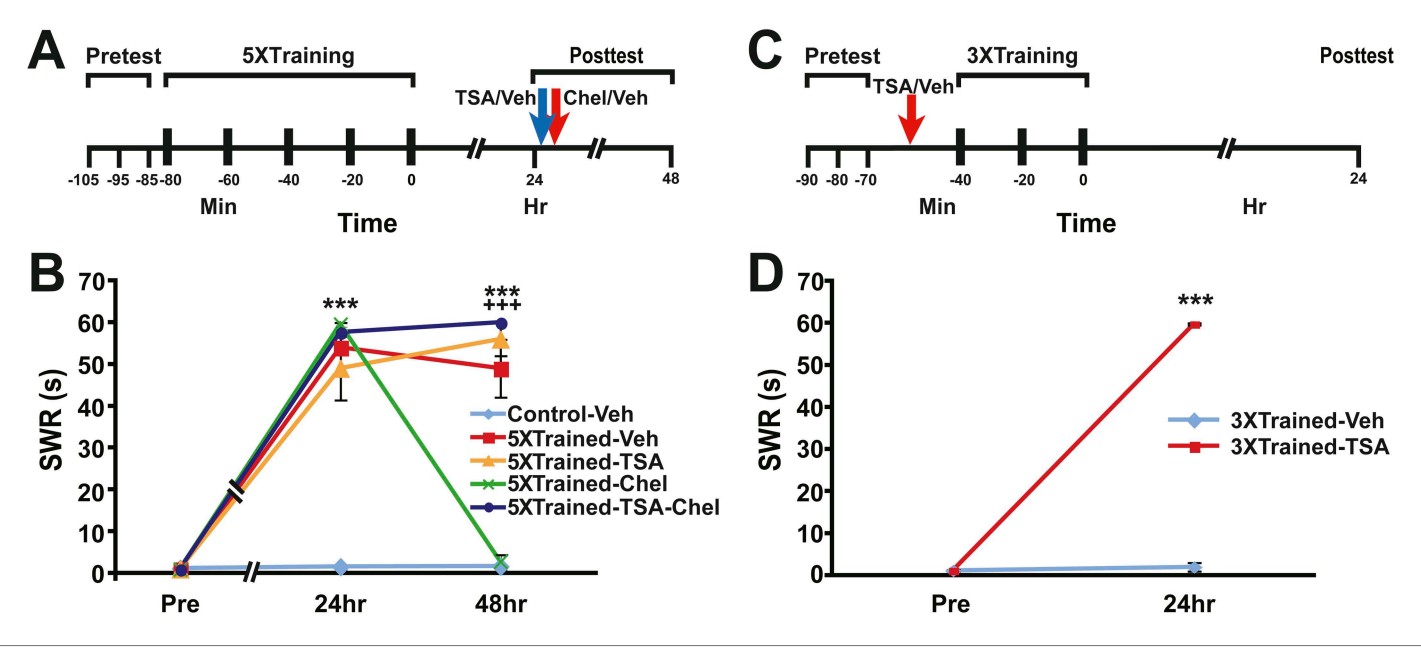

**Figure 9**. Epigenetic regulation of LTM in *Aplysia*. (**A**) Experimental protocol. The times of the pretests, training, posttests, and drug/vehicle injections are shown relative to the end of the last training session. The times of the trichostatin A (TSA)/vehicle and chelerythrine/vehicle injections are indicated by blue arrow and red arrow respectively. (**B**) TSA treatment blocks erasure of LTS by chelerythrine. The graph presents the mean duration of the SWR measured at 24 hr and 48 hr in the Control-Veh (n = 9), 5XTrained-Veh (n = 5), 5XTrained-TSA (n = 6), 5XTrained-Chel (n = 5), and 5XTrained-TSA-Chel (n = 6) groups. A repeated-measures ANOVA showed a significant group × time interaction ($F_{(8,52)}$ = 53.2, p < 0.0001). Subsequent planned comparisons indicated that the group differences for the 24 and 48 hr posttests were highly significant (24 hr, $F_{[4,26]}$ = 45.5, p < 0.0001; 48 hr, $F_{[4,26]}$ = 94.4, p < 0.0001). For the 24 hr posttest, SNK posthoc tests indicated that the training produced significant sensitization in all four trained groups (5XTrained-Veh, 54.0 ± 6.0 s; 5XTrained-TSA, 49.0 ± 7.6 s; 5XTrained-Chel, 59.6 ± 0.4 s; and 5XTrained-TSA-Chel, 57.7 ± 2.3 s) compared to the Control-Veh group (1.6 ± 0.6 s, p < 0.001 for all tests). Comparisons of the four trained groups showed that their responses did not differ significantly on the 24 hr posttest. However, the responses of 5XTrained-Veh (49.0 ± 6.9 s), 5XTrained-TSA (56.0 ± 4.0 s), and 5XTrained-TSA-Chel (59.8 ± 1.7 s) on the 48 hr posttest were significantly more enhanced than those for both the 5XTrained-Chel (2.8 ± 1.8 s) and Control-Veh (1.7 ± 0.5 s) groups (p < 0.001 for all tests). Thus, chelerythrine treatment reversed LTS, and TSA treatment blocked chelerythrine's reversal of LTS. There were no significant differences among 5XTrained-Veh, 5XTrained-TSA and 5XTrained-TSA-Chel groups, nor between 5XTrained-Chel and Control-Veh groups, at 48 hr. Asterisks, comparisons of the Control-Veh group with the 5XTrained-Veh group, the 5XTrained-TSA group, the 5XTrained-Chel group, and the 5XTrained-TSA-Chel group at 24 hr; and of the Control-Veh group with the 5XTrained-Veh group, the 5XTrained-TSA group, and the 5XTrained-TSA-Chel group at 48 hr. Plus signs, comparisons of the 5XTrained-Chel group with the 5XTrained-Veh group, the 5XTrained-TSA group, and the 5XTrained-TSA-Chel group at 48 hr. (**C**) Experimental protocols. The times of the pretests, training, posttests, and drug/vehicle injections are shown relative to the end of the last training session. The intrahemocoel injection of either TSA or vehicle is indicated by the red arrow. Note that TSA was injected into animals prior to the sensitization training. (**D**) Inhibiting histone deacetylation facilitated the induction of LTS in Aplysia. The mean duration of the SWR in the 3XTrained-TSA group (n = 4) at 24 hr was 59.8 ± 0.3 s, whereas it was only 2.0 ± 1.0 s in the 3XTrained-Veh group (n = 4; p < 0.001, unpaired *t* test). Asterisks, comparison between 3XTrained-TSA and 3XTrained-Veh groups; Error bars represent ±SEM.

explanation incorporates the notion of an as-yet-unidentified priming mechanism; here, reconsolidation blockade and inhibition of PKM, can erase the stored sensitization memory through the reversal of pre- and postsynaptic structures induced by the long-term training, but the antimnemonic treatments do not eliminate the primer. The primer does not constitute LTM, but is required for its reconstitution via new synaptic growth. The priming signal might interact with fresh facilitatory input to the withdrawal circuit, due to the additional (3×) tail shocks, to upregulate the number of synaptic contacts; the appropriate number of contacts could be determined by the homeostatic process, and involve signals from existing and new synapses.

In a previous study we attempted to reinstate LTS following its disruption by inhibition of PKM Apl III but failed (*Cai et al., 2011*) (see *Figure 3B* of the earlier study). Interestingly, we used only a single bout of tail shocks in that attempt. In light of the present results (*Figures 7 and 8*), our previous failure indicates that one bout of tails shocks is insufficient to fully recruit the signaling pathways that reinstate LTS (*Abel et al., 1998*; *Sossin, 2008*; *Mayford et al., 2012*; *Zhang et al., 2012*). In future work

we will seek to discover how the molecular signals evoked by three bouts of tail shocks differ from those evoked by a single bout.

The data from our behavioral experiments involving the use of an HDAC inhibitor (*Figure 9*) implicate epigenetic modifications (*Levenson and Sweatt 2005*; *Rahn et al., 2013*; *Graff and Tsai 2013a*)—either in the SN or MN or, most likely, both—in the storage mechanism for LTM in *Aplysia*. Data from an in vitro study of LTF have also implicated histone acetylation in LTM in *Aplysia* (*Guan et al., 2002*). How might HDAC inhibition block chelerythrine's apparent disruption of LTM? Possibly, TSA, by inhibiting histone deacetylase, activates the transcription of genes that promote LTM; the activation of these genes, in turn, may compensate for a disruptive action of chelerythrine on LTM. Findings by others are consistent with this idea. Thus, treatment with an HDAC inhibitor rescues cognitive deficits and learning-related synaptic plasticity in mice that are heterozygous for a mutant form of CREB binding protein (CBP) (*Alarcon et al., 2004*). In addition, it has been found that phosphorylation of CBP by the atypical PKCζ is required for the histone acetylation that promotes the differentiation of developing neural precursors into neurons, astrocytes, and oligodendrocytes; mutant mice haploinsufficient for CBP exhibit abnormal development of the cerebral cortex (*Wang et al., 2010*). Furthermore, chelerythrine, by inhibiting the phosphorylation of CBP by PKCζ, blocks the differentiation of cultured precursor cells into astrocytes, and oligodendrocytes (*Wang et al., 2010*). A recent study in *Aplysia* has found that reducing the activity of CBP in SNs through intracellular injection of CBP small interfering RNA (siRNA) impaired the induction of LTF in sensorimotor cocultures (*Liu et al., 2013*). Interestingly, this deficit in LTF induction could be rescued through the use of a 5HT training protocol designed to maximize the phosphorylation and synthesis of CCAAT/enhancer-binding protein (C/EBP). Taken together, these studies suggest that chelerythrine can disrupt epigenetic processes, possibly involving histone acetylation stimulated by CBP, that promote the induction and maintenance of LTM. Moreover, because HDAC inhibition can block chelerythrine's deleterious effects on memory, as well as promote the induction of LTM, it is intriguing to speculate that these effects might be due to enhancement of a CBP-dependent pathway.

But the above explanation for why TSA blocks chelerythrine's effect on consolidated LTM raises a critical question. Our data suggest that consolidated LTM persists after chelerythrine treatment (as well as reconsolidation blockade), possibly as nuclear changes. If so, then the molecular basis for the persistence of LTM is unlikely to be histone acetylation, because an apparent action of chelerythrine is to reverse histone acetylation. Another prominent epigenetic mechanism known to play a role in LTM is DNA methylation (*Day and Sweatt, 2010*). Although early evidence indicated that learning-induced DNA methylation in the hippocampus was transient and readily reversible (*Miller and Sweatt, 2007*), a more recent study has reported that contextual fear conditioning in rats induces DNA methylation of the gene for calcineurin in cortical neurons that persists for at least a month (*Miller et al., 2010*). Furthermore, Sweatt and colleagues have shown in a rat model of childhood maltreatment that early trauma can produce changes in the DNA methylation of the gene for brain-derived neurotrophic factor (BDNF) in the cortex, changes that persist into adulthood (*Roth et al., 2009*). Thus, DNA methylation may constitute an epigenetic mechanism for the lifelong storage of memory (*Day and Sweatt, 2010*). Interestingly, 5HT has recently been reported to induce DNA methylation of the promoter of the transcriptional repressor of memory CREB2, thereby facilitating the induction of LTF (*Rajasethupathy et al., 2012*). Finally, another potential candidate for a nuclear mechanism of LTM storage is suggested by a recent study by *Suberbielle et al. (2013)* that found, remarkably, that learning causes DNA double-strand breaks (DSBs) in neurons in the brains of control mice; thus, chromatin remodeling subsequent to DNA DSBs may also encode LTM.

Could a nonsynaptic storage mechanism based on nuclear changes mediate the maintenance of associative memories, particularly those induced in complex neural circuits in the mammalian brain, where a given neuron may have 1000s or 10s of 1000s of synaptic partners? An obvious difficulty confronting any hypothetical nuclear storage mechanism in the mammalian brain is how the appropriate number of connections can be maintained in a synapse-specific manner after learning has occurred. For example, if a synaptic contact that has undergone Hebbian long-term potentiation (*Bliss and Collingridge, 1993*) as a consequence of associative learning retracts, how could a nuclear storage mechanism restrict the growth of a replacement contact to the correct pair of pre- and postsynaptic neurons? Possibly, there are nonsynaptic ways for neurons to communicate that ensure specificity of associative synaptic plasticity in the face of the significant lability of synaptic structure documented here.

There are, of course, somal, non-nuclear, mechanisms of memory storage or priming that could account for the apparent persistence of memory following synaptic erasure. One such mechanism would be the persistent activity of a kinase (*Hegde et al., 1993*; *Bougie et al., 2012*), or persistent inhibition of a phosphatase (*Sharma et al., 2003*), in the cell body of the SN or MN or in both cell bodies. Another somal mechanism consistent with our data is ubiquitination of one or more somal proteins (*Hegde et al., 1997*; *Chain et al., 1999*); up-regulation of a ubiquitin-proteasome pathway could degrade proteins that inhibit the storage of LTM, or that block the priming of LTM.

The present data necessitate a reappraisal of the mnemonic consequences of blockade of memory reconsolidation and inhibition of PKM. It has been previously argued that these manipulations can erase consolidated memory (*Nader and Hardt, 2009*; *Sacktor, 2011*; *Glanzman, 2013*) (although see *Lattal and Abel, 2004*). But our results indicate that the effect of reconsolidation blockade and PKM inhibition is not to delete LTM but, rather, to impair its expression. In other words, the antimnemonic actions of these manipulations may result from their ability to at least partially reverse the cellular and molecular changes, including synaptic growth, that mediate the expression, rather than the storage, of LTM. Possibly, neither reconsolidation blockade nor PKM inhibition can reverse the nuclear remodeling—involving epigenetic modifications, particularly DNA methylation (*Day and Sweatt, 2010*; *Graff et al., 2011*; *Rahn et al., 2013*), as well as, perhaps, chromatin remodeling subsequent to DNA DSBs (*Suberbielle et al., 2013*)—that represents the stored memory trace. Alternatively, these antimnemonic manipulations might indeed disrupt the stored memory trace, but leave intact some memory-priming signal. Likely candidates for a residual priming signal include the persistent repression of transcription of the CREB repressor, CREB2 (*Bartsch et al., 1995*; *Upadhya et al., 2004*; *Liu et al., 2011*), and persistent degradation of the regulatory subunit of protein kinase A (PKA) (*Chain et al., 1999*).

Because chelerythrine's disruption of LTM in *Aplysia* is blocked by TSA, chelerythrine's antimnemonic actions must involve histone deacetylation. Previous speculation regarding how chelerythrine and the zeta inhibitory peptide (ZIP) disrupt memory maintenance in both *Aplysia* and rats has focused on the ability of these pharmacological agents to inhibit the phosphorylation of proteins by atypical PKM (*Cai et al., 2011*; *Sacktor, 2011*) (*Box 1*). The present results imply that chelerythrine and ZIP may interfere with epigenetic processes induced by atypical PKM, as well as inhibit protein phosphorylation by atypical PKM. A prior study reported that, following inhibition of PKMζ activity in the insular cortex of rats by the peptide inhibitor ZIP, conditioned taste aversion could not be reinstated by an application of the unconditioned stimulus (UCS, intraperitoneal injection of lithium chloride) (*Shema et al., 2007*). Our experience (*Cai et al., 2011*, and the present results) suggests that the failure to reinstate conditioned taste aversion in this study was not because LTM had been extinguished, but because a single application of the UCS was too weak to reverse the disruptive effects of ZIP on mechanisms of LTM expression, some of which may involve chromatin remodeling (*Figure 7A,B*).

In summary, we have found that LTM, or a primer for LTM, can persist following reconsolidation blockade and inhibition of PKM. Furthermore, the residual memory/primer must be independent of synaptic plasticity, because it persists following elimination of the synaptic changes induced during learning. Our results indicate that consolidated memories may be far more refractory to modification or elimination than generally supposed. If confirmed and extended to mammals, the present results would have important implications for treating disorders of LTM, such as posttraumatic stress disorder (PTSD).

## Materials and methods

### Cell culture and dye injection

The sensorimotor cocultures consisted of one pleural SN and one small siphon (LFS-type) MN. Adult abdominal ganglia and pleural ganglia were removed from 60–100 g *Aplysia* and then incubated in protease (10 mg/ml Dispase II [Roche Applied Science, Indianapolis, IN] in Leibowitz-15 [L-15, Sigma, St Louis, MO]) for 2 hr at 35°C before desheathing. The appropriate amounts of salts were added to the L15 to yield the following concentrations in mM: 400 NaCl, 11 CaCl$_2$, 10 KCl, 27 MgSO$_4$, 27 MgCl$_2$, 2 NaHCO$_3$. Additionally, the L15 was supplemented with penicillin (50 unit/ml), streptomycin (50 μg/ml), dextrose (6 mg/ml) and glutamine (0.1 mg/ml). After desheathing, SNs and MNs were individually dissociated from ganglia and paired in cell culture. The culture medium contained 50% *Aplysia* hemolymph and 50% L-15. The cultures were maintained at 18°C in an incubator for 3 day before the start

## Box 1. Recent studies of transgenic mice have reported that learning and memory, as well as long-term synaptic plasticity in the hippocampus, are normal in animals lacking PKMζ, the mammalian homolog of PKC Apl III (Lee et al., 2013; Volk et al., 2013).

Although these results raise questions about the necessity of PKMζ for maintaining long-term memory in the mammalian brain, as well as the specificity of the peptide inhibitor (ZIP) commonly used to block the activity of PKMζ in mammalian studies (Ling et al., 2002; Pastalkova et al., 2006; Shema et al., 2007), we do not believe they impact the present conclusions. Biochemical tests performed in Aplysia have established that chelerythrine, at the concentration used in our experiments, is a selective and effective inhibitor of PKM (Villareal et al., 2009). Furthermore, 5HT has been shown to activate PKM Apl III via calpain dependent cleavage of the atypical Aplysia PKC, PKC Apl III (Bougie et al., 2012). A possible explanation for the results of the PKMζ knockout mice (Lee et al., 2013; Volk et al., 2013), which lack the gene for the atypical PKCζ (note that in the mammalian CNS PKMζ is not formed from proteolytic cleavage of the atypical PKC, but, rather, is transcribed from an alternate promoter within the PKCζ gene [Hernandez et al., 2003]) is that, in the absence of PKMζ, a second atypical mammalian PKC isoform, PKCι/λ—or its PKM fragment, PKMι/λ—can assume the mnemonic functions of PKMζ (Ren et al., 2013). But there is only one atypical PKC isoform in Aplysia; we therefore believe that chelerythrine's inhibitory effect in the present experiments was selective for PKM Apl III.

of the experiments. The SNs and MNs were labeled with the intracellular dyes dextran fluorescein and dextran rhodamine B (Molecular Probes, Eugene, OR), respectively, on Day 3/4 in culture. The dyes were dissolved in 0.2 M KCl with 0.25% fast green (final concentration of 10 mg/ml) and then microinjected into cells via brief pressure pulses (6–12 MΩ resistance electrodes, 10–20 psi for 2–5 pulses [40 ms]).

### Cell imaging and quantification of structural changes

The fluorescent images of the labeled cocultures were acquired with a LSM Pascal (Zeiss, Thornwood, NY) confocal microscope during three imaging sessions: immediately prior to, and 24 hr and 48 hr after, 5X5HT training (or at the equivalent times for Control cocultures). The images were taken using a 20×, 0.5 NA objective. The total number of varicosities was counted for each sensory neuron from the confocal images. All image analyses were performed using Axiovision 4.8.2 (Zeiss, Thornwood, NY). The counter was blind to the experimental conditions of the imaged cocultures. SN varicosities that were in clear contact with, or overlapping, postsynaptic structures (either the MN soma, major neurite, or fine processes) were counted. The majority of the SN varicosities contacted either the MN soma or its initial segment. Only those varicosities having a punctate shape—those in which an oval-shaped body could be distinguished by the narrowing of the neurite on either side—and a measured area of 10 μm² or greater were counted. If a large fluorescent varicosity comprised several visible punctate varicosities in the image, those small varicosities were counted individually; if not, the structure was treated as a single varicosity. Three categories of varicosities are tracked and quantified: original varicosities, that is the varicosities present at 0 hr; varicosities formed during 0–24 hr; and varicosities formed during 24–48 hr.

### Drug treatments

Immediately after the first imaging session, some of the cocultures were given 5HT training. 5HT was prepared fresh daily as a 10 mM stock solution in artificial sea water (ASW) and then diluted to the final concentration of 100 μM in the perfusion medium immediately before the first application. The perfusion solution contained 50% L15 and 50% ASW. The 5HT training consisted of five

5 min pulses of 5HT. After each 5 min pulse, the 5HT was rapidly washed out with normal perfusion medium for 15 min. The Control cocultures were treated with the perfusion solution alone. Following 5HT or control treatment, the perfusion medium was replaced with culture medium and the cocultures were returned to the 18°C incubator. A stock solution of anisomycin (Sigma, St Louis, MO) was made by dissolving the protein synthesis inhibitor in dimethyl sulfoxide (DMSO) to a concentration of 40 mM for use in the reconsolidation experiments. After the second imaging session (24 hr after 5HT training or the equivalent time in Control experiments) the anisomycin stock solution was diluted to a concentration of 10 µM in perfusion solution and applied to cocultures for 2 hr. Immediately prior to the anisomycin treatment, some cocultures were given one 5-min pulse of 5HT (100 µM) as the reminder stimulus. The 5HT-containing perfusion medium was washed out with normal perfusion medium in the cocultures that received the reminder. To inhibit PKM Apl III chelerythrine (EMD Biosciences, San Diego, CA) was dissolved in dH$_2$O to a concentration of 10 mM to make a stock solution. After the second imaging session 24 hr after 5HT training, the chelerythrine stock solution was diluted to a concentration of 10 µM in perfusion solution and applied to cocultures for 1 hr. Following the anisomycin or chelerythrine treatment, the drug was rapidly washed out of the coculture dishes with normal perfusion medium; afterwards the perfusion medium was replaced with culture medium and the cocultures were returned to the incubator.

## Behavioral experiments

Adult *A. californica* (80–120 g) were obtained from a local supplier (Alacrity Marine Biological, Redondo Beach, CA, USA). Animals were housed in a 50 gal aquarium filled with cooled (12–14°C), aerated seawater (Catalina Water Company, Long Beach, CA, USA). The behavioral training and testing methods were similar to those previously described (*Fulton et al., 2008*; *Cai et al., 2011*, *2012*). Three pretests were performed at once per 10 min, beginning 25 min before the start of training. Full sensitization training consisted of five bouts of electrical shocks delivered to the tail at 20-min intervals. During each bout, the animal received three trains of shocks spaced 2 s apart. Each train was 1 s in duration; the shocks (10-ms pulse duration, 40 Hz, 120 V) were delivered via a Grass stimulator (S88, Astro-Med, West Warwick, RI) connected to platinum wires implanted in the tail. After training the animals were given posttests as indicated in the figures.

A stock solution of 40 mM anisomycin or 10 mM chelerythrine was prepared as for the cell imaging experiments (above). Anisomycin was then diluted in ASW to a concentration of 20 mM (50% DMSO). 200 µl per 100 g of body weight of anisomycin was injected into the animals. Injections of the same amount of vehicle solution (DMSO in ASW) were made in Control experiments. The final concentrations of anisomycin in the animal were approximately 40 µM. The final concentration of DMSO in the hemocoel was ~0.1%. A volume of 200 µl per 100 g of body weight of chelerythrine was injected into the animals. Trichostatin A (TSA) (Sigma, St Louis, MO) was dissolved in DMSO to a concentration of 10 mM to make a stock solution. To inhibit the histone deacetylase, a volume of 100 µl per 100 g of body weight of TSA was injected into the animals. The specific times at which the intrahemocoel injections were made are indicated in the relevant figures.

## Statistical analysis

All statistical tests were performed using SPSS 22.0 (IBM, Armonk, NY). Parametric tests were used for all statistical analyses. For each coculture the number of varicosities counted at 24 hr and 48 hr after 5HT training were normalized as described in the figure legends. The normalized data were expressed as means ±SEM. For the analysis of the data involving monitoring of synapses/behavior over time, the overall data were first assessed with repeated-measures two-way analyses of variance (ANOVA). If the repeated-measures ANOVA indicated a significant interaction, one-way ANOVAs were performed on the separate test times, followed by Student-Newman-Keuls (SNK) posthoc tests for pairwise comparisons. An unpaired Student's *t*-test was used to determine the statistical significance of the differences when there were only two groups in the data set (data in *Figures 2,5,6 and 9D*). All reported levels of significance represent two-tailed values unless otherwise indicated.

## Acknowledgements

This study was supported by research grants to DLG from the National Institute of Neurological Disorders and Stroke (NIH R01 NS029563), the National Institute of Mental Health (NIH R01 MH096120), and the National Science Foundation (IOS 1121690). We thank R Singh, M Kimbrough, K Sarmiento,

X Zhao and T Dehghani for assistance with the behavioral training, and MS Fanselow, FB Krasne, and WS Sossin for helpful comments on the manuscript.

## Additional information

### Funding

| Funder | Grant reference number | Author |
|---|---|---|
| National Institute of Neurological Disorders and Stroke | R01 NS029563 | David L Glanzman |
| National Institute of Mental Health | R01 MH096120 | David L Glanzman |
| Division of Integrative Organismal Systems | IOS 1121690 | David L Glanzman |

The funders had no role in study design, data collection and interpretation, or the decision to submit the work for publication.

### Author contributions

SC, Conception and design, Acquisition of data, Analysis and interpretation of data; DC, DLG, Conception and design, Analysis and interpretation of data, Drafting or revising the article; KP, PY-WS, Acquisition of data, Analysis and interpretation of data; ACR, Analysis and interpretation of data

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
