## [Decision Letter]

Thank you for sending your work entitled “Reinstatement of long-term memory following erasure of its behavioral and synaptic expression in *Aplysia*” for consideration at *eLife*. Your article has been favorably evaluated by K VijayRaghavan (Senior editor), a Reviewing editor, and two reviewers.

The following individuals responsible for the peer review of your submission have agreed to reveal their identity: Mani Ramaswami (Reviewing editor); Tom Abrams and Ron Calabrese (peer reviewers).

The Reviewing editor and the other reviewers discussed their comments before we reached this decision, and the Reviewing editor has assembled the following comments to help you prepare a revised submission.

The manuscript contains a very interesting set of results on mechanisms of long-term synaptic plasticity obtained in a classic system for investigating the cellular and molecular basis of long-term memory: the sensory neuron to motor neuron synapse from Aplysia reconstituted in culture. Most significantly, the experiments performed here show that: (a) procedures that reverse LTS in Aplysia sensorimotor co-cultures leads to loss of synaptic varicosities, which occurs without the selective retraction of LTS-induced synaptic varicosities; (b) LTM can be reinstated in intact animals after its erasure by training that will not induce LTM in naïve animals.

These observations make the major point that the persistence of memory is not determined by the stability of synapses. Instead it depends on a different persistent cellular mechanism that drives synaptic change in order to enable the expression of memory. In other words, memory storage is independent of synaptic change, but memory expression requires synaptic change. In other words, without synaptic change memory exists but is covert. This intellectual and didactic point, which is broadly supported by the authors' results, merits publication in *eLife*.

However, in current form the manuscript does not do a good job of positioning their findings within the literature on long-term synaptic plasticity spanning the past 2 decades. A major careful and attentive rewrite is required, which acknowledges previous observations and clarifies how previous observations and ideas may be integrated with and reinterpreted in authors' new perspective.

The language used in the current manuscript has the potential to confound and confuse. For instance the statement that “LTM is independent of synaptic change in Aplysia” is certain to confuse. What is meant is that memory is covertly encoded and stored, possibly in the cytoplasm and possibly in the nucleus; this memory signal induces synaptic changes necessary for memory expression. For a point as subtle, specific and interesting as this, loose statements such as “memory is independent of synaptic change” are confusing as well as unproductively provocative. The exaggeration of the radical nature of these results sometimes make them appear unnecessarily paradoxical, potentially reducing their impact.

In any case, the authors should rewrite, clarify and qualify statements and conclusions in the manuscript in way that tries to bring the rest of the field around to the author's point of view. Many specific suggestions and arguments are offered below. A particularly strong recommendation, elaborated in point 2, is to consider presenting the new perspective in the framework of a memory priming model, in which non synaptic priming mechanisms may be long-lasting and store memory.

Several are stylistic, intellectual or theoretical points with which the authors should engage constructively in a revised discussion. These should be viewed as constructive and supportive suggestions that the reviewers believe will improve the presentation and impact of the manuscript. It is not necessary that the authors should accept all of these suggestions. However, a revised manuscript should list the authors’ response to each suggestion.

1) Assigning memory storage to a change in the nucleus is not unreasonable, but yet it is entirely speculative. This is particularly true when the apparent deacetylation of histones triggered by chelerythrine does not compromise the covert persistent memory. Logically, this means that the residual memory that persists after PKC inhibition and which can be fully restored (or unmasked) with 3 more training trials, cannot be mediated by histone acetylation. This is an important point of confusion in the authors' analysis. Although they wish to assign a role for histone acetylation in the “synapse-independent” memory mechanism, their evidence actually suggests that there is an additional mechanism of memory beyond the PKC-dependent memory, the erasure of which is blocked by TSA. This issue should be more objectively and critically discussed.

2) Priming vs. occult memory. The authors should acknowledge that it is possible that the original sequence of 5-HT exposures primes the synapses for plasticity even after a memory is erased. There is a substantial number of experimental results suggesting priming, both at the synaptic and the behavioral levels (e.g. [33]). The Barco et al (2002, Figure 5) result with constitutively active CREB suggests changes in gene expression can subsequently interact with local synaptic signals to initiate synaptic plasticity. Even the Guan et al. result with an HDAC inhibitor that enables a single pulse of 5-HT to initiate long-term facilitation suggests there synapses can be primed for long-term plasticity. Thus, early training can induce molecular changes (e.g. transcription of an IEG or translocation to the nucleus) but subsequent training may be required to induce the change in synaptic strength or behavior. These priming mechanisms are considered to be molecular explanations for the importance of repeated training trials. If the Chen-Glanzman phenomenon worked similarly, there could be a persistent change that primes cells to respond to subsequent training trials. Histone acetylation could be one such change, though the TSA data are not consistent with this possibility. A change in an ubiquitination-proteasome protein or in a kinase or phosphatase are examples of non-genomic persistent changes that could contribute. Seen from this perspective, the synaptic changes would be part of the memory, but there are cell wide changes that would be more persistent. This is a less paradoxical suggestion, but one that could be equally interesting. It has the advantage of preserving the possibility of synapse specific plasticity contributing to learning. The authors should acknowledge and discuss these possibilities.

3) In these experiments, the stability of both basal and long-term facilitated synaptic strength over days suggests a homeostatic mechanism that specifies net synaptic strength, rather than constraining the strength of individual synaptic connections. There seems to be a set point for net synaptic strength, although this set point may shift during long-term synaptic plasticity (much as Hu and colleagues have described.) That the overall strength of synaptic connections may be kept stable while individual synaptic connections form or are retracted does not necessarily mean that synapses don't “store memory.” Instead, it suggests the possibility that there is a mechanism to set global synaptic strength, much as suggested by numerous studies of homeostatic synaptic scaling. The concept of a set point that shifts with sensitization should be discussed more explicitly.

4) From a molecular perspective, although it is tempting to assign primary responsibility for memory representation to a single protein or gene and to focus attention on this single player (at least within a given study), this is clearly not a valid way to understand the biology of memory. All molecular participants in memory storage interact with a wide array of other molecules in complex networks. We know from studies such as Casadio et al. (1999) and Barco et al. (2002) that changes in transcription interact with local synaptic “tags” to initiate long-term alterations in synaptic strength. Indeed, the emergent conclusion over nearly two decades is that neither the transcriptional changes nor local synaptic changes are sufficient to represent long-term memory. The authors suggest that synapses merely express memory that is “stored” at the genome level. However, they present no evidence that the homeostatic set point is specified solely in the genome (or even the transcriptome), rather than emerging from a dynamic process in which synaptic sites participate interactively in storing the memory. To consider a crude analogy, this is a bit like positing that muscle strength is specified in muscle fiber nuclei, rather than through complex dynamic interactions among gene expression, cytoskeletal elements, signaling molecules and extracellular matrix, and also non-muscle cells, simply because individual myosin or actin filaments turn over.

The discussion may be better received if the role of the synapse in directing synapse specific memory at least, were acknowledged and integrated (for instance, see point 5 below).

5) The authors should discuss the possibility that synapses are merely toggled between a basal state and a facilitated state (with 2 amplitudes of connections), such that the apparent restoration of a covert memory is actually just a switching back to the facilitated, which occurs more readily after priming. There may be good evidence that this is not the case, but it should be explicitly considered.

6) In terms of placing the present study in a broader context, it would be helpful for the authors to discuss studies on interactions between transcription and local signals, such as those mentioned above, and also recent studies on mammalian CNS that demonstrate remarkable morphological stability of individual dendritic spines over weeks or months, e.g. by Yi Zuo and colleagues.

7) One of the most novel results in this manuscript is the finding in the final Figure that the reversal of long-term memory by transient inhibition of atypical protein kinase C (homologous with PKM zeta in mammalian brain) requires histone deactylase activity. This result requires reevaluation of the long-argued model that persistent activity of PKM is required for memory because it supports enhanced synaptic strength at individual local synaptic sites. Even with this one data set, these results might have a larger impact in a separate short publication, rather than tacked on at the end of the present study. This perspective is partially supported by the authors' conclusions, which rather trivialize this important finding. (For example in the Abstract, it is stated: “we provide evidence that LTM in Aplysia is stored by epigenetic changes” and nearly identical summary statements appear at the end of the Introduction and in the Discussion.) We have known for more than a decade in both Aplysia and mammalian CNS that histone acetylation is critical for long-term synaptic plasticity and long-term memory. What is novel here is that atypical PKC activity appears to be required for the persistence of the histone acetylation-dependent effect, possibly by suppressing histone deactylase activity.

8) An important prediction of the hypothesis that synaptic growth is an expression mechanism of LTM not LTM itself that in co-cultures one should be able to re-express more synaptic growth after 5X HT reminder-anisomycin 3X 5HT than in naive 3X 5HT. Can this be tested and quantified? (Perhaps these data are already available?)

9) Although paper is generally clearly written, it is very tedious to read because of the complicated nature of the protocols. This may simply be impossible to avoid but the authors can try even harder to make the writing flow. Further, the paper depends heavily on corresponding physiological studies in co-cultures, and the authors should be at great pains to point out the details of these studies (e.g., Cai D, Pearce K, Chen S, Glanzman DL. 2011. Protein kinase M maintains long-term sensitization and long-term facilitation in Aplysia. J. Neurosci. 31: 6421-31. Cai D, Pearce K, Chen S, Glanzman DL. 2012. Reconsolidation of long-term memory in Aplysia. Curr. Biol. 22: 1783-88) in relevant regions of the text.

10) It is particularly intriguing that memory reversal restored the number of presynaptic varicosities to precisely the number before memory formation. This point could be interpreted more vigorously as it fits the prior model in which synapse plasticity underlies memory, and could form the basis then for the authors' intellectual reinterpretation (that memory exists though it is invisible when extra synapses are lost).

11) Statistics. Although the authors perform ANOVAs on their data, the ANOVA results are not presented. Simply including posthoc tests is not appropriate as the overall ANOVA result is as important as the posthoc test (arguably more so). F values and exact p values should be included for each ANOVA.

12) Even if the authors wish to combine the 5x5-HT without a 5-HT reminder group with the 5x5-HT without a 5-HT reminder but with late anisomycin into a single group for purposes of statistical comparison with other treatment groups, we should be shown the data for the two treatments separately, at least in the initial figure (Figure 1).

13) Abstract. “Further, we show that LTS can be reinstated following its apparent elimination by reconsolidation blockade and inhibition of PKM, treatments that erase LTS-related synaptic growth.” This sentence fails to communicate that memory persists covertly after apparent elimination. Reinstatement suggests only that relearning is possible.

14) The term “+Reminder group” for the synapses that receive a reminder of 5-HT plus anisomycin is a poor term, as there is no suggestion of the use of a protein synthesis inhibitor. In Figure 7, this same group is called 5xTrained-Reminder-Aniso. In the legend of Figure 2 it states “Coculture in which synaptic reconsolidation was disrupted (+Reminder coculture),” which is somewhat confusing. It's not the reminder 5-HT that disrupts memory but the reminder + inhibition of protein synthesis.

15) From the Introduction, the sentence “These morphological results imply that LTM is independent of synaptic change in Aplysia” seems internally inconsistent. Clearly the authors believe that memory involves synaptic modification as they quantify the number of synaptic contacts as a read out for memory.

16) Results. The reversal of morphological change with anisomycin after 5-HT “supports the idea that the reminder triggered reconsolidation of the synaptic growth that mediates LTF.” Reconsolidation is what occurs with a reminder alone (e.g. 5-HT alone). Anisomycin treatment blocks the reconsolidation. This should be stated more clearly, particularly in a journal with broad readership. This could be explained in the Introduction when reconsolidation is first mentioned.

17) Results: “there was also significant retraction of the original varicosities during this period in both groups of trained sensorimotor coculture, those that received the reminder stimulus and those that did not.” The description fails to mention that retraction of pre-existing varicosities was significantly greater in the Reminder + anisomycin group. This merits emphasis.

18) Results: “retraction observed previously in the 5-HT group.” Probably the authors are referring to the group that received the 5-HT reminder plus anisomycin (most of the groups received 5-HT).

19) Results: “If this idea is correct, then it should be possible to dissociate LTM in Aplysia from synaptic facilitation.” This should be clarified.

20) Results: “as indicated by comparing … without the reminder training” This detailed description of the groups to be compared is not necessary in Results. The sentence up to this point made it clear that the three additional 5-HT exposures restored the erased memory.

21) Results: “three bouts of tail shocks can reinstate LTM following the suppression of its behavioral expression” This sentence is rather inappropriate for the Results, as it is a bit circular. The authors can logically suggest that inhibition of atypical PKC suppresses the expression of long-term memory, rather than erasing it, only because the subsequent tail shocks reinstate (or reestablish) the memory. Certainly, the evidence in this manuscript is consistent with a covert memory persisting after blockade of reconsolidation and inhibition of PKM. However, this conclusion about suppression, rather than erasure, of memory should actually be considered in the Discussion.

22) Results: “the synaptic mechanism the mediates” should have “that” before “mediates.”

23) Discussion: “Could a nonsynaptic, epigenetic storage mechanism, such as the one we propose for Aplysia, mediate the maintenance of associative memories, particularly those induced in complex neural circuits in the mammalian brain, where a given neuron may have thousands or tens of thousands of synaptic partners? This may seem unlikely.” Do the authors really mean to suggest that the mechanism their data suggests is relevant only for simpler invertebrate nervous systems?

24) L-15, Discussion: Was this medium supplemented with salts?

25) Discussion: “oval-shaped main body” This is an odd description of a varicosity, as it suggests that there are regions of the varicosity outside the “main body.”

26) Discussion: “unpaired Student's t-test.” When was the t-test used instead of an ANOVA?

27) In the experiments of Figures 1 and 7, it seems very important to this reviewer to have the control of a reminder stimulus (after training - 5X 5HT) without the anisomycin, i.e. 5X Trained Reminder. It seems even more important than the naïve Control because we want to know that the reminder itself has no effect. Am I wrong here? Can the authors explain why this control is not important? The legend for Figure 1 actually suggests this control was done (“A reminder pulse of 5-HT (single blue bar) was delivered to the +Reminder cocultures prior to the anisomycin/vehicle”).

28) Figure 3 statistical comparison. Because the authors have performed ANOVAs, it is possible to compare the amount of retraction of newly formed varicosities with the retraction of original varicosities (e.g. Figures 3 and 5). These results should be included.

---

## [Author Response]

*1) Assigning memory storage to a change in the nucleus is not unreasonable, but yet it is entirely speculative. This is particularly true when the apparent deacetylation of histones triggered by chelerythrine does not compromise the covert persistent memory. Logically, this means that the residual memory that persists after PKC inhibition and which can be fully restored (or unmasked) with 3 more training trials, cannot be mediated by histone acetylation. This is an important point of confusion in the authors' analysis. Although they wish to assign a role for histone acetylation in the “synapse-independent” memory mechanism, their evidence actually suggests that there is an additional mechanism of memory beyond the PKC-dependent memory, the erasure of which is blocked by TSA. This issue should be more objectively and critically discussed*.

We agree with this important point and we thank the reviewers for pointing out our error. In our discussion of the potential role of epigenetic mechanisms we now acknowledge that the chelerythrine results argue against the idea that the storage mechanism for LTM involves histone acetylation. We now propose an alternative epigenetic mechanism for LTM storage, DNA methylation. As we discuss, recent evidence from David Sweatt’s lab have shown that DNA methylation changes, induced by learning or by early trauma, may persist for the life of an animal. Chromatin remodeling subsequent to DNA double-strand breaks (71) is another potential nuclear mechanism for LTM storage.

*2) Priming vs. occult memory. The authors should acknowledge that it is possible that the original sequence of 5-HT exposures primes the synapses for plasticity even after a memory is erased. There is a substantial number of experimental results suggesting priming, both at the synaptic and the behavioral levels (e.g.*
[33]*,*
[34]*). The Barco et al (2002,*
Figure 5*) result with constitutively active CREB suggests changes in gene expression can subsequently interact with local synaptic signals to initiate synaptic plasticity. Even the Guan et al. result with an HDAC inhibitor that enables a single pulse of 5-HT to initiate long-term facilitation suggests there synapses can be primed for long-term plasticity. Thus, early training can induce molecular changes (e.g. transcription of an IEG or translocation to the nucleus) but subsequent training may be required to induce the change in synaptic strength or behavior. These priming mechanisms are considered to be molecular explanations for the importance of repeated training trials. If the Chen-Glanzman phenomenon worked similarly, there could be a persistent change that primes cells to respond to subsequent training trials. Histone acetylation could be one such change, though the TSA data are not consistent with this possibility. A change in an ubiquitination-proteasome protein or in a kinase or phosphatase are examples of non-genomic persistent changes that could contribute. Seen from this perspective, the synaptic changes would be part of the memory, but there are cell wide changes that would be more persistent. This is a less paradoxical suggestion, but one that could be equally interesting. It has the advantage of preserving the possibility of synapse specific plasticity contributing to learning. The authors should acknowledge and discuss these possibilities*.

We are grateful to the reviewers for the insightful discussion. We have rewritten the manuscript to incorporate the possibility that our results reflect the persistence of a priming mechanism required to reestablish the disrupted LTM, rather than LTM itself. In addition, we explicitly compare the priming explanation for our results to the idea that LTM was not erased by reconsolidation blockade/chelerythrine treatment.

*3) In these experiments, the stability of both basal and long-term facilitated synaptic strength over days suggests a homeostatic mechanism that specifies net synaptic strength, rather than constraining the strength of individual synaptic connections. There seems to be a set point for net synaptic strength, although this set point may shift during long-term synaptic plasticity (much as Hu and colleagues have described.) That the overall strength of synaptic connections may be kept stable while individual synaptic connections form or are retracted does not necessarily mean that synapses don't “store memory.” Instead, it suggests the possibility that there is a mechanism to set global synaptic strength, much as suggested by numerous studies of homeostatic synaptic scaling. The concept of a set point that shifts with sensitization should be discussed more explicitly*.

We have added an explicit discussion of the notion that the overall strength of synaptic connections is regulated by a homeostatic mechanism that toggles between a facilitated state and a nonfacilitated state, as determined by experience. However, our argument against the idea that synapses store memory does not rest simply on the idea that the overall strength of a sensorimotor connection remains stable while individual synapses grow and retract. More powerful evidence against the notion of synaptic storage comes from our finding that when the homeostatic mechanism resets the total number of synaptic connections to the pretraining value, the resulting morphology is quite different from the original one, and that memory can be reinstated in the animal following antimnemonic treatments that erase learning-related synaptic growth.

*4) From a molecular perspective, although it is tempting to assign primary responsibility for memory representation to a single protein or gene and to focus attention on this single player (at least within a given study), this is clearly not a valid way to understand the biology of memory. All molecular participants in memory storage interact with a wide array of other molecules in complex networks. We know from studies such as Casadio et al. (1999) and Barco et al. (2002) that changes in transcription interact with local synaptic “tags” to initiate long-term alterations in synaptic strength. Indeed, the emergent conclusion over nearly two decades is that neither the transcriptional changes nor local synaptic changes are sufficient to represent long-term memory. The authors suggest that synapses merely express memory that is “stored” at the genome level. However, they present no evidence that the homeostatic set point is specified solely in the genome (or even the transcriptome), rather than emerging from a dynamic process in which synaptic sites participate interactively in storing the memory. To consider a crude analogy, this is a bit like positing that muscle strength is specified in muscle fiber nuclei, rather than through complex dynamic interactions among gene expression, cytoskeletal elements, signaling molecules and extracellular matrix, and also non-muscle cells, simply because individual myosin or actin filaments turn over*.

*The discussion may be better received if the role of the synapse in directing synapse specific memory at least, were acknowledged and integrated (for instance, see point 5 below)*.

LTM induction may not be a useful model for LTM storage. In particular, although the results of Barco et al. and those of others indicate that CREB-dependent changes in gene expression can interact with local synaptic tags to initiate LTF, as the reviewer states, our results do not provide any evidence for synaptic tagging. This is because the retraction of sensory neuron varicosities following reconsolidation blockade or chelerythrine treatment appeared arbitrary. The notion of synaptic tagging during LTM storage is difficult to reconcile with this apparently arbitrary elimination of varicosities, both original and 5HT-induced. Also, it seems to us that the homeostatic mechanism, whatever it is, resides in either the nucleus or the cell body; the reviewer’s idea that synaptic homeostasis results from a dynamic interaction between some priming mechanism and labeled synaptic sites only seems viable in the context of synaptic tagging. It is possible that the synapses that remain following reconsolidation blockade or chelerythrine treatment contain a 5HT-induced molecular signal (the tag), which is why they survive antimnemonic treatment; this tag may interact with the residual priming mechanism in the nucleus or cell body to restore LTM.

*5) The authors should discuss the possibility that synapses are merely toggled between a basal state and a facilitated state (with 2 amplitudes of connections), such that the apparent restoration of a covert memory is actually just a switching back to the facilitated, which occurs more readily after priming. There may be good evidence that this is not the case, but it should be explicitly considered*.

As pointed out in the response to Comment 3 above, we agree with the reviewers that implicit in our data is the possibility that “synapses are merely toggled between a basal state and a facilitated state”. (Note that there must also be a third state to incorporate the evidence that the sensorimotor synapse can exhibit long-term depression.) But the idea of an experience-dependent toggling between basal and facilitated states is ultimately consistent with either a persistent priming mechanism or covert memory.

*6) In terms of placing the present study in a broader context, it would be helpful for the authors to discuss studies on interactions between transcription and local signals, such as those mentioned above, and also recent studies on mammalian CNS that demonstrate remarkable morphological stability of individual dendritic spines over weeks or months, e.g. by Yi Zuo and colleagues*.

The revised manuscript now includes a discussion of this important issue. Briefly, significant controversy exists regarding the extent to which spines are stable over long periods of time in the mammalian brain. Also, even in those studies in which spines have been reported to be mostly stable after learning, there is a large degree of spine turnover in the first few days after training, which is when our measurements were made. So our results are not necessarily contradictory with those of in vivo mammalian imaging studies.

*7) One of the most novel results in this manuscript is the finding in the final Figure that the reversal of long-term memory by transient inhibition of atypical protein kinase C (homologous with PKM zeta in mammalian brain) requires histone deactylase activity. This result requires reevaluation of the long-argued model that persistent activity of PKM is required for memory because it supports enhanced synaptic strength at individual local synaptic sites. Even with this one data set, these results might have a larger impact in a separate short publication, rather than tacked on at the end of the present study. This perspective is partially supported by the authors' conclusions, which rather trivialize this important finding. (For example in the Abstract, it is stated: “we provide evidence that LTM in Aplysia is stored by epigenetic changes” and nearly identical summary statements appear at the end of the Introduction and in the Discussion.) We have known for more than a decade in both Aplysia and mammalian CNS that histone acetylation is critical for long-term synaptic plasticity and long-term memory. What is novel here is that atypical PKC activity appears to be required for the persistence of the histone acetylation-dependent effect, possibly by suppressing histone deactylase activity*.

We agree with the reviewers that this finding is novel and important, and we certainly do not wish to trivialize it by including it here. We believe the finding properly belongs in the paper because it supports our general position that nuclear changes are central to the persistence of long-term memory (although we agree with the reviewer that histone acetylation cannot account for the reinstatement of memory following chelerythrine treatment). The fact that TSA blocks chelerythrine’s disruptive effect on LTM also echoes the other finding in Figure 9, namely that TSA can convert the intermediate-term sensitization induced by three bouts of tail shocks into LTS. The revised Discussion includes an analysis of the role of PKC activity in epigenetic regulation of LTM induction and maintenance in *Aplysia*. Also, we have rewritten the Abstract, Introduction and Discussion to eliminate the redundant statements cited by the reviewers.

*8) An important prediction of the hypothesis that synaptic growth is an expression mechanism of LTM not LTM itself that in co-cultures one should be able to re-express more synaptic growth after 5X HT reminder-anisomycin 3X 5HT than in naive 3X 5HT. Can this be tested and quantified? (Perhaps these data are already available?*)

Although our results do predict this result, testing it would require a significant effort. Notice that to be convincing the morphological experiments proposed by the reviewer would have to be preceded by electrophysiological experiments confirming that the 5X5HT-1X5HT-Aniso-3X5HT treatment produces LTF and that, as we would expect, 3X5HT alone does not. These experiments have not yet been done. Also, because the experiments would require intracellular electrophysiological recording from pre- and postsynaptic neurons of the cocultures for three consecutive days, their rate of success would be low. I estimate that the electrophysiological and morphological experiments necessary to test the prediction would take my lab the better part of a year to complete. Therefore, I hope the reviewers will agree to allow us to publish without these data.

*9) Although paper is generally clearly written, it is very tedious to read because of the complicated nature of the protocols. This may simply be impossible to avoid but the authors can try even harder to make the writing flow. Further, the paper depends heavily on corresponding physiological studies in co-cultures, and the authors should be at great pains to point out the details of these studies (e.g., Cai D, Pearce K, Chen S, Glanzman DL. 2011. Protein kinase M maintains long-term sensitization and long-term facilitation in Aplysia. J. Neurosci. 31: 6421-31. Cai D, Pearce K, Chen S, Glanzman DL. 2012. Reconsolidation of long-term memory in Aplysia. Curr. Biol. 22: 1783-88) in relevant regions of the text*.

I have worked to streamline the writing in the paper, but the experimental protocols are admittedly not simple. Regarding the reviewer’s other point, the results of our previous electrophysiological studies are detailed in the relevant sections of the text so that the reader can see precisely how our present morphological results correspond to the prior synaptic physiological results.

*10) It is particularly intriguing that memory reversal restored the number of presynaptic varicosities to precisely the number before memory formation. This point could be interpreted more vigorously as it fits the prior model in which synapse plasticity underlies memory, and could form the basis then for the authors' intellectual reinterpretation (that memory exists though it is invisible when extra synapses are lost)*.

We agree with the reviewers that this is an intriguing result and have emphasized it in several places in the manuscript. The result is explicitly discussed in the second paragraph of the Discussion, where we point out that the restoration of the number of presynaptic varicosities to the pretraining level following reconsolidation blockade/chelerythrine treatment is paralleled by the results from our prior electrophysiological studies (13, 14). The parallel between our results and those of studies of homeostatic synaptic plasticity in other systems is also mentioned here.

*11) Statistics. Although the authors perform ANOVAs on their data, the ANOVA results are not presented. Simply including posthoc tests is not appropriate as the overall ANOVA result is as important as the posthoc test (arguably more so). F values and exact p values should be included for each ANOVA*.

The results for all ANOVAs in our study, including the exact F- and p-values for each test, are presented in the relevant figure legends.

*12) Even if the authors wish to combine the 5x5-HT without a 5-HT reminder group with the 5x5-HT without a 5-HT reminder but with late anisomycin into a single group for purposes of statistical comparison with other treatment groups, we should be shown the data for the two treatments separately, at least in the initial figure (*Figure 1*)*.

Figure 1 now presents the data for the 5X5HT and 5X5HT-Aniso groups graphed separately, as requested.

*13) Abstract: “Further, we show that LTS can be reinstated following its apparent elimination by reconsolidation blockade and inhibition of PKM, treatments that erase LTS-related synaptic growth.” This sentence fails to communicate that memory persists covertly after apparent elimination. Reinstatement suggests only that relearning is possible*.

The Abstract has been rewritten. It now states, “In addition, we find evidence that the LTM for sensitization persists covertly after its apparent elimination by the antimnemonic treatments that erase learning-related synaptic growth.”

*14) The term “+Reminder group” for the synapses that receive a reminder of 5-HT plus anisomycin is a poor term, as there is no suggestion of the use of a protein synthesis inhibitor. In*
Figure 7*, this same group is called 5xTrained-Reminder-Aniso. In the legend of*
Figure 2
*it states “Coculture in which synaptic reconsolidation was disrupted (+Reminder coculture),” which is somewhat confusing. It's not the reminder 5-HT that disrupts memory but the reminder + inhibition of protein synthesis*.

The reviewers are correct; the original group labels were misleading. We have replaced them with new terms that we hope will be both less confusing and more accurate. Thus, the former “– Reminder” group has now been separated into two experimental groups for the graph in Figure 1, the “5X5HT” and 5X5HT-Aniso” groups. The former “+Reminder” group is now labeled the “5X5HT-1X5HT-Aniso” group. The new group labels in Figure 1 now more closely parallel those in Figure 7. In addition, the use of the new labels eliminates the inaccuracy pointed out by the reviewer for the legend in Figure 2.

Notice that we have also used new group labels for Figure 3. The former “–Reminder” group (which comprises the combined data for the 5X5HT, 5X5HT-1X5HT and 5X5HT-Aniso groups) is now more accurately labeled, “No reconsolidation/No blockade”. The former “+Reminder” group is now labeled the “5X5HT-1X-5HT-Aniso” group, as in Figure 1.

*15) From the Introduction, the sentence “These morphological results imply that LTM is independent of synaptic change in Aplysia” seems internally inconsistent. Clearly the authors believe that memory involves synaptic modification as they quantify the number of synaptic contacts as a read out for memory*.

This sentence has been rewritten as follows: “These results imply that the persistence of memory does not require the stability of particular synaptic connections.”

*16) Results. The reversal of morphological change with anisomycin after 5-HT “supports the idea that the reminder triggered reconsolidation of the synaptic growth that mediates LTF.” Reconsolidation is what occurs with a reminder alone (e.g. 5-HT alone). Anisomycin treatment blocks the reconsolidation. This should be stated more clearly, particularly in a journal with broad readership. This could be explained in the Introduction when reconsolidation is first mentioned*.

The sentence has been revised, and now states, “This structural result…provides additional support for the notion that 1X5HT reactivated the synaptic memory induced by the 5X5HT training”. In addition, the Introduction now includes an explanation of the phenomenon of memory reconsolidation.

*17) Results: “there was also significant retraction of the original varicosities during this period in both groups of trained sensorimotor coculture, those that received the reminder stimulus and those that did not.” The description fails to mention that retraction of pre-existing varicosities was significantly greater in the Reminder + anisomycin group. This merits emphasis*.

We have added the following statement to the Results: “Varicosities in the 5X5HT-1X5HT-Aniso group—the group of trained cocultures subjected to reconsolidation blockade—exhibited a similar pattern of retraction of 5HT-induced and original varicosities, but the amount of retraction was significantly greater than that observed in the No reconsolidation/No blockade cocultures. Note that “5X5HT-1X5HT-Aniso” is the new label for the former “+ Reminder” group and “No reconsolidation/No blockade” is the new label for the former “–Reminder” group (see the response to Comment 14 above).

18) Results: “retraction observed previously in the 5-HT group.” Probably the authors are referring to the group that received the 5-HT reminder plus anisomycin (most of the groups received 5-HT).

We apologize for this confusing sentence. We have revised it to make our meaning clear: “Monitoring of the fate of individual varicosities in the 5X5HT group revealed the same pattern of structural growth and retraction observed previously in the 5X5HT-trained cocultures not subjected to reconsolidation blockade.”

*19) Results: “If this idea is correct, then it should be possible to dissociate LTM in Aplysia from synaptic facilitation.” This should be clarified*.

This sentence has been eliminated. The introduction to this section of the paper has been rewritten to make our meaning clearer. The first paragraph now reads:

“The present morphological results challenge the notion that the persistence of sensitization memory in *Aplysia* depends on the persistence of particular facilitated synapses. To further investigate this idea, we tested whether the LTM for behavioral sensitization can be reinstated in *Aplysia* following reconsolidation blockade and inhibition of PKM, two treatments previously shown to eliminate LTF, the synaptic basis of long-term sensitization…”

*20) Results: “as indicated by comparing...... without the reminder training” This detailed description of the groups to be compared is not necessary in Results. The sentence up to this point made it clear that the three additional 5-HT exposures restored the erased memory*.

We have eliminated this detailed description.

*21) Results: “three bouts of tail shocks can reinstate LTM following the suppression of its behavioral expression” This sentence is rather inappropriate for the Results, as it is a bit circular. The authors can logically suggest that inhibition of atypical PKC suppresses the expression of long-term memory, rather than erasing it, only because the subsequent tail shocks reinstate (or reestablish) the memory. Certainly, the evidence in this manuscript is consistent with a covert memory persisting after blockade of reconsolidation and inhibition of PKM. However, this conclusion about suppression, rather than erasure, of memory should actually be considered in the Discussion*.

This statement has been removed.

*22) Results: “the synaptic mechanism the mediates” should have “that” before “mediates*.*”*

The sentence containing this mistake has been eliminated.

*23) Discussion: “Could a nonsynaptic, epigenetic storage mechanism, such as the one we propose for Aplysia, mediate the maintenance of associative memories, particularly those induced in complex neural circuits in the mammalian brain, where a given neuron may have thousands or tens of thousands of synaptic partners? This may seem unlikely.” Do the authors really mean to suggest that the mechanism their data suggests is relevant only for simpler invertebrate nervous systems*?

We appreciate the reviewers’ question. No, we do not believe that our data are only relevant to invertebrate nervous systems. We have eliminated these statements in the Discussion.

*24) L-15, Discussion: Was this medium supplemented with salts*?

It is now stated in the Material and methods that the L-15 was supplemented with the appropriate salts.

*25) Discussion: “oval-shaped main body” This is an odd description of a varicosity, as it suggests that there are regions of the varicosity outside the “main body*.*”*

The descriptive term “oval-shaped main body” has been changed to “oval-shaped body”.

*26) Discussion: “unpaired Student's t-test” When was the t-test used instead of an ANOVA*?

The summary of our statistical methods now states, “An unpaired Student’s *t*-test was used to determine the statistical significance of the differences when there were only two groups in the data set such as the data presented in Figures 3, 5, 6, 8 and 9.”

*27) In the experiments of*
Figures 1 and 7*, it seems very important to this reviewer to have the control of a reminder stimulus (after training - 5X 5HT) without the anisomycin, i.e. 5X Trained Reminder. It seems even more important than the naïve Control because we want to know that the reminder itself has no effect. Am I wrong here? Can the authors explain why this control is not important? The legend for*
Figure 1
*actually suggests this control was done (“A reminder pulse of 5-HT (single blue bar) was delivered to the +Reminder cocultures prior to the anisomycin/vehicle”)*.

We did include this control in our morphological experiments; it is referred to as the “5X5HT-1X5HT” group. A supplemental graph for Figure 1 (Figure 1—figure supplement 1) is included in the revised MS; it presents the mean overall varicosity number, normalized to the number of varicosities present at 0 hr, for each of the three groups of trained cocultures not subjected to reconsolidation blockade.

One-way ANOVAs performed on the varicosity data for the 24hr and 48hr tests confirmed that the differences among the groups were insignificant. Notice that the 5X5HT and 5X5HT-1X5HT groups were combined into one group (labeled the “5X5HT” group) for the graph in Figure 1. Also, the three groups shown in Figure 1—figure supplement 1, the 5X5HT, 5X5HT-1X5HT and 5X5HT-Aniso groups, were combined into the No reconsolidation/No blockade group in Figure 3.

We did not include this control in the behavioral experiments.

*28)*
Figure 3
*statistical comparison. Because the authors have performed ANOVAs, it is possible to compare the amount of retraction of newly formed varicosities with the retraction of original varicosities (e.g.*
Figures 3 and 5*). These results should be included*.

These results are included in the revised MS as Figure 3—figure supplement 1 and Figure 5—figure supplement 1. An interesting result revealed by the graphs in these new figures is that, whereas the antimnemonic treatments—reconsolidation blockade and chelerythrine treatment—resulted in approximately equal retraction of 5-HT-induced and original varicosities, there was significantly greater retraction of 5-HT-induced varicosities than of original varicosities in the 5X5HT-trained groups not subjected to either antimnemonic treatment.